# Multi-agent cooperation through learning-aware policy gradients

**Alexander Meulemans**[1,*]**, Seijin Kobayashi**[1,*]**, Johannes von Oswald**[1]**, Nino Scherrer**[1]**,
Eric Elmoznino**[1,2,3]**, Blake A. Richards**[1,2,3,4,5]**, Guillaume Lajoie**[1,2,3,4,5]**,
Blaise Agüera y Arcas**[1]**, João Sacramento**[1]

[1]Google, Paradigms of Intelligence Team, [2]Mila - Quebec AI Institute, [3]Université de Montréal,
[4]McGill University, [5]CIFAR, [*]Equal contribution
ameulemans@google.com, seijink@google.com

## Abstract

Self-interested individuals often fail to cooperate, posing a fundamental challenge for multi-agent learning. How can we achieve cooperation among self-interested, independent learning agents? Promising recent work has shown that in certain tasks cooperation can be established between "learning-aware" agents who model the learning dynamics of each other. Here, we present the first unbiased, higher-derivative-free policy gradient algorithm for learning-aware reinforcement learning, which takes into account that other agents are themselves learning through trial and error based on multiple noisy trials. We then leverage efficient sequence models to condition behavior on long observation histories that contain traces of the learning dynamics of other agents. Training long-context policies with our algorithm leads to cooperative behavior and high returns on standard social dilemmas, including a challenging environment where temporally-extended action coordination is required. Finally, we derive from the iterated prisoner's dilemma a novel explanation for how and when cooperation arises among self-interested learning-aware agents.

## 1 Introduction

From self-driving autonomous vehicles to personalized assistants, there is a rising interest in developing agents that can learn to interact with humans (Collins et al., 2024; Gweon et al., 2023), and with each other (Park et al., 2023; Vezhnevets et al., 2023). However, multi-agent learning comes with significant challenges that are not present in more conventional single-agent paradigms. This is perhaps best seen through the study of "social dilemmas", general-sum games which model the tension between cooperation and competition in abstract form (von Neumann & Morgenstern, 1947). Without further assumptions, letting agents independently optimize their individual objectives on such games results in poor outcomes and a lack of cooperation (Tan, 1993; Claus & Boutilier, 1998).

First, for general-sum games, reaching an equilibrium point does not necessarily imply appropriate behavior because there can be many sub-optimal equilibria (Fudenberg & Levine, 1998; Shoham & Leyton-Brown, 2008). Second, the control problem an agent faces is non-stationary from its own viewpoint, because other agents themselves simultaneously learn and adapt (Hernandez-Leal et al., 2017). Centralized training algorithms sidestep non-stationarity issues by sharing agent information (Sunehag et al., 2017), but this transformation into a global learning problem is usually prohibitively costly, and impossible to implement when agents must be developed separately (Zhang et al., 2021).

The above two fundamental issues have hindered progress in multi-agent reinforcement learning, and have limited our understanding of how self-interested agents may reach high returns when faced with social dilemmas. In this paper, we join a promising line of work on "learning awareness" that has been shown to improve cooperation (Foerster et al., 2018a). The key idea behind such approaches is to take into account the learning dynamics of other agents explicitly, rendering it into a meta-learning problem (Schmidhuber, 1987; Bengio et al., 1990; Hochreiter et al., 2001).

The present paper contains two main novel results on learning awareness in general-sum games. First, we introduce a new learning-aware reinforcement learning rule derived as a policy gradient estimator. Unlike existing methods (Foerster et al., 2018a;b; Xie et al., 2021; Balaguer et al., 2022;

Lu et al., 2022; Willi et al., 2022; Cooijmans et al., 2023; Khan et al., 2024; Aghajohari et al., 2024), it has a number of desirable properties: (i) it does not require computing higher-order derivatives, (ii) it is provably unbiased, (iii) it can model minibatched learning algorithms, (iv) it is applicable to scalable architectures based on recurrent sequence policy models, and (v) it does not assume access to privileged information, such as the opponents' policies or learning rules. Our policy gradient rule significantly outperforms previous model-free methods in the general-sum game setting. In particular, we show that efficient learning-aware learning suffices to reach cooperation in a challenging sequential social dilemma involving temporally extended actions (Leibo et al., 2017) that we adapt from the Melting Pot suite (Agapiou et al., 2023). Second, we analyze the iterated prisoner's dilemma (IPD), a canonical model for studying cooperation among self-interested agents (Rapoport, 1974; Axelrod & Hamilton, 1981). Our analysis uncovers a novel mechanism for the emergence of cooperation through learning awareness, and explains why the seminal learning with opponent-learning awareness algorithm due to Foerster et al. (2018a) led to cooperation in the IPD.

## 2 BACKGROUND AND PROBLEM SETUP

We consider partially observable stochastic games (POSGs; Kuhn, 1953) consisting of a tuple $(\mathcal{I}, \mathcal{S}, \mathcal{A}, P_t, P_r, P_i, \mathcal{O}, P_o, \gamma, T)$ with $\mathcal{I} = \{1, \ldots, n\}$ a finite set of $n$ agents, $\mathcal{S}$ the state space, $\mathcal{A} = \times_{i \in \mathcal{I}} \mathcal{A}^i$ the joint action space, $P_t(S_{t+1} \mid S_t, A_t)$ the state transition distribution, $P_i(S_0)$ the initial state distribution, $P_r = \times_{i \in \mathcal{I}} P_r^i(R \mid S, A)$ the joint factorized reward distribution with $R = \{R^i\}_{i \in \mathcal{I}}$ and bounded rewards $R^i$, $\mathcal{O} = \times_{i \in \mathcal{I}} \mathcal{O}^i$ the joint observation space, $P_o(O_t \mid S_t, A_{t-1})$ the observation distribution, $\gamma$ the discount factor, $t$ the time step, and $T$ the horizon. We use superscript $i$ to indicate agent-specific actions, observations and rewards, $-i$ to indicate all agent indices except $i$, and we omit the superscript for joint actions, observations and rewards. As agents only receive partial state information, they benefit from conditioning their policies $\pi^i(a_t^i \mid x_t^i; \phi^i)$ on the observation history $x_t^i = \{o_k^i\}_{k=1}^t$ (Åström, 1965; Kaelbling et al., 1998), with $\phi^i$ the policy parameters. Note that the observations can contain the agent's actions on previous timesteps.

### 2.1 GENERAL-SUM GAMES AND THEIR CHALLENGES

We focus on general-sum games, where each agent has their own reward function, possibly different from those of other agents. Specifically, we consider mixed-motive general-sum games that are neither zero-sum nor fully-cooperative. Analyzing and solving such general-sum games while letting every agent individually and independently maximize their rewards (a setting often referred to as "fully-decentralized reinforcement learning"; Albrecht et al., 2024) is a longstanding problem in the fields of machine learning and game theory for two primary reasons, described below.

**Non-stationarity of the environment.** In a general-sum game, each agent aims to maximize its expected return $J^i(\phi^i, \phi^{-i}) = \mathbb{E}_{P^{\phi^i, \phi^{-i}}} \left[ \sum_{t=1}^T \gamma^t R_t^i \right]$, with $P^{\phi^i, \phi^{-i}}$ the distribution over environment trajectories $x_T$ induced by the environment dynamics, the policy $\pi^i(a^i \mid x^i; \phi^i)$ of agent $i$, and the policies $\pi^{-i}$ of all other agents. Importantly, the expected return $J^i(\phi^i, \phi^{-i})$ does not only depend on the agent's own policy, but also on the current policies of the other agents. As other agents are updating their policies through learning, the environment which includes the other agents is effectively non-stationary from a single agent's perspective. Furthermore, the actions of an agent can influence this non-stationarity by changing the observation histories of other agents, on which they base their learning updates.

**Equilibrium selection.** It is not clear how to identify appropriate policies for a general-sum game. To see this, let us first briefly revisit the concept of a Nash equilibrium (Nash Jr., 1950). For a fixed set of co-player policies $\phi^{-i}$, one can compute a best response, which for agent $i$ is given by $\phi^{\star i} = \arg\max_{\phi^i} J^i(\phi^i, \phi^{-i})$. When all current policies $\phi$ are a best response against each other, we have reached a Nash equilibrium, where no agent is incentivized to change its policy anymore, $\forall i, \tilde{\phi}^i : J^i(\tilde{\phi}^i, \phi^{-i}) \leq J^i(\phi^i, \phi^{-i})$. Various "folk theorems" show that for most POSGs of decent complexity, there exist infinitely many Nash equilibria (Fudenberg & Levine, 1998; Shoham & Leyton-Brown, 2008). This lies at the origin of the equilibrium selection problem in multi-agent reinforcement learning: it is not only important to let a multi-agent system converge to a Nash equilibrium, but also to target a *good* equilibrium, as Nash equilibria can be arbitrarily bad. Famously, unconditional mutual defection in the infinitely iterated prisoner's dilemma is a Nash equilibrium, with strictly lower expected returns for all agents compared to the mutual tit-for-tat Nash equilibrium (Axelrod & Hamilton, 1981).

## 2.2 CO-PLAYER LEARNING AWARENESS

We aim to address the above two major challenges of multi-agent learning in this paper. Our work builds upon recent efforts that are based on adding a meta level to the multi-agent POSG, where the higher-order variable represents the learning algorithm used by each agent (Lu et al., 2022; Balaguer et al., 2022; Khan et al., 2024). In this meta-problem, the environment includes the learning dynamics of other agents. At the meta-level, one episode now extends across multiple episodes of actual game play, allowing the "ego agent", $i$, to observe how its co-players, $-i$, learn, see Fig. 1. The goal of this meta-agent may be intuitively understood as that of *shaping* co-player learning to its own advantage.

Provided that co-player learning algorithms remain constant, the above reformulation yields a single-agent problem that is amenable to standard reinforcement learning techniques. This setup is fundamentally asymmetric: while the *meta agent* (ego agent) is endowed with co-player learning awareness (i.e., observing multiple episodes of game play), the remaining agents remain oblivious to the fact that the environment is non-stationary. We thus refer to them here as *naive agents* (see Fig. 1B). Despite this asymmetry, prior work has observed that introducing a learning-aware agent in a group of naive learners often leads to better learning outcomes for all agents involved, avoiding mutual defection equilibria (Lu et al., 2022; Balaguer et al., 2022; Khan et al., 2024). Moreover, Foerster et al. (2018a) has shown that certain forms of learning awareness can lead to the emergence of cooperation even in symmetric cases, a surprising finding that is not yet well understood.

These observations motivate our study, leading us to derive novel efficient learning-aware reinforcement learning algorithms, and to investigate their efficacy in driving a group of agents (possibly composed of both meta and naive agents) towards more beneficial equilibria. Below, we proceed by first formalizing asymmetric co-player shaping problems, which we solve with a novel policy gradient algorithm (Section 3). In Section 4, we then return to the question of why and when co-player learning awareness can result in cooperation in multi-agent systems with equally capable agents.

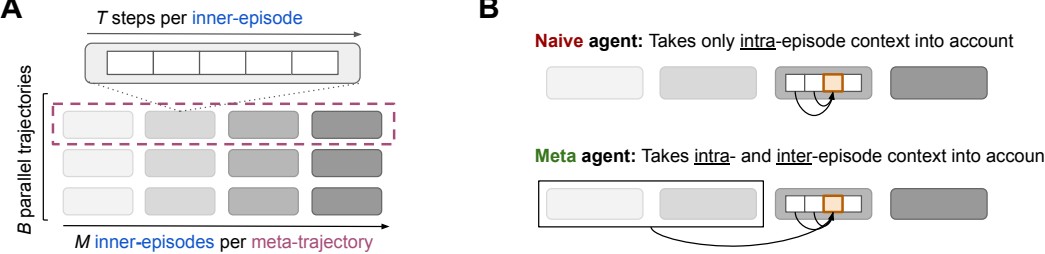

**Figure 1: A.** Experience data terminology. Inner-episodes comprise $T$ steps of (inner) game play, played between agents $B$ times in parallel, forming a batch of inner-episodes. A given sequence of $M$ inner-episodes forms a meta-trajectory, thus comprising $MT$ steps of inner game play. The collection of $B$ meta-trajectories forms a meta-episode. **B.** During game play, a naive agent takes only the current episode context into account for decision making. In contrast, a meta agent takes the full long context into account. Seeing multiple episodes of game play endows a meta agent with learning awareness.

**Co-player shaping.** Following Lu et al. (2022), we first introduce a meta-game with a single meta-agent whose goal is to shape the learning of naive co-players to its advantage. This meta-game is defined formally as a single-agent partially observable Markov decision process (POMDP) $(\tilde{\mathcal{S}}, \tilde{\mathcal{A}}, \tilde{P}_t, \tilde{P}_r, \tilde{P}_i, \tilde{\mathcal{O}}, \tilde{\gamma}, M)$. The meta-state consists of the policy parameters $\phi^{-i}$ of all co-players together with the agent's own parameters $\phi^i$. The meta-environment dynamics represent the fixed learning rules of the co-players, and the meta-reward distribution represents the expected return $J^i(\phi^i, \phi^{-i})$ collected by agent $i$ during an inner episode, with "inner" referring to the actual game being played. The initialization distribution $\tilde{P}_i$ reflects the policy initializations of all players. Finally, we introduce a meta-policy $\mu(\phi^i_{m+1} \mid \phi^i_m, \phi^{-i}_m; \theta)$ parameterized by $\theta$, that decides the update to the parameter $\phi^i_{m+1}$ (the meta-action) to shape the co-player learning towards highly rewarding regions for agent $i$ over a horizon of $M$ meta steps. This leads to the co-player shaping problem

$$\max_{\mu} \mathbb{E}_{\tilde{P}_i(\phi^i_0, \phi^{-i}_0)} \mathbb{E}_{\tilde{P}^\mu} \left[ \sum_{m=1}^{M} J^i(\phi^i_m, \phi^{-i}_m) \right], \tag{1}$$

with $\tilde{P}^\mu$ the distribution over parameter trajectories induced by the meta-dynamics and meta-policy.

## 2.3 Single-level co-player shaping by leveraging sequence models

In this paper, we combine both inner- and meta-policies in a single long-context policy, conditioning actions on long observation histories spanning multiple inner game episodes (see Fig. 1B). Instead of hand-designing the co-player learning algorithms, we instead let meta-learning discover the algorithms used by other agents. This way, we leverage the in-context learning and inference capabilities of modern neural sequence models (Brown et al., 2020; Rabinowitz, 2019; Akyürek et al., 2023; von Oswald et al., 2023; Li et al., 2023) to both simulate in-context an inner policy, as well as strategically update it based on current estimates of co-player policies. This philosophy has been adopted in Khan et al. (2024), in which a flat policy is optimized using an evolutionary algorithm. We compare to this method in Section 3.1, after we derive our meta reinforcement learning algorithm.

To proceed with this approach, we must first reformulate the meta-game. In particular, we must deal with a difficulty that is not present in single-agent meta reinforcement learning (e.g., Wang et al., 2016; Duan et al., 2017), which stems from the fact that co-players generally update their own policies based on multiple inner episodes ("minibatches"), without which reinforcement learning cannot practically make progress. Here, we solve this by defining the environment dynamics over $B$ parallel trajectories, with $B$ the size of the minibatch of inner episode histories that co-players use to update their policies at each inner episode boundary (see Fig. 1A).

**Batched co-player shaping POMDP.** We define the batched co-player shaping POMDP $(\bar{\mathcal{S}}, \bar{\mathcal{A}}, \bar{P}_t, \bar{P}_r, \bar{P}_i, \bar{\mathcal{O}}, \bar{\gamma}, M, B)$, with hidden states consisting of the hidden environment states of the $B$ ongoing inner episodes, combined with the current parameters $\phi_m^{-i}$ of all co-players; environment dynamics $\bar{P}_t$ simulating $B$ environments in parallel, combined with updating the co-player's policy parameters $\phi^{-i}$, and resetting the environments at each inner episode boundary; initial state distribution $\bar{P}_i$ that initializes the co-player policies and initializes the environments for the first inner episode batch; and finally, an ego-agent policy $\bar{\pi}^i(\bar{a}_l^i \mid \bar{h}_l^i; \phi^i)$ parameterized by $\phi^i$, which determines a distribution over the batched action $\bar{a}_l^i = \{a_l^{i,b}\}_{b=1}^B$, based on the batched long history $\bar{h}_l^i = \{h_l^{i,b}\}_{b=1}^B$. We refer to each element of the latter as a long history $h_l^{i,b}$, with long time index $l$ running across multiple episodes, from $l=1$ until $l=MT$. It should be contrasted to the inner episode history $x_t^i$, which runs from $t=1$ to $t=T$ and thus only reflects the current (inner) game history.

The POMDP introduced above suggests using a sequence policy $\bar{\pi}^i(\bar{a}_l^i \mid \bar{h}_l^i; \phi^i)$ that is aware of the full minibatch of long histories and which produces a joint distribution over all current actions in the minibatch. However, as we aim to use our agents not only to shape naive learners, but also to play against/with each other, we require a policy that can be used both in a batch setting with naive learners, and in a single-trajectory setting with other learning-aware agents. Within our single-level approach, we achieve this by factorizing the batch-aware policy $\bar{\pi}^i(\bar{a}_t^i \mid \bar{h}_l^i; \phi^i)$ into $B$ independent policies with shared parameters $\phi^i$, $\bar{\pi}^i(\bar{a}_l^i \mid \bar{h}_l^i; \phi^i) = \prod_{b=1}^B \pi^i(a_l^{i,b} \mid h_l^{i,b}; \phi^i)$. Thanks to the batched POMDP, we can now pose co-player shaping as a standard (single-level, single-agent) expected return maximization problem:

$$\max_{\phi^i} \mathbb{E}_{\bar{P}^{\phi^i}} \left[ \frac{1}{B} \sum_{b=1}^B \sum_{l=0}^{MT} R_l^{i,b} \right]. \tag{2}$$

This formulation is the key for obtaining an efficient policy gradient co-player shaping algorithm.

## 3 Co-agent learning-aware policy gradients

### 3.1 A policy gradient for shaping naive learners

We now provide a meta reinforcement learning algorithm for solving the co-player shaping problem stated in Eq. 2 efficiently. Under the POMDP introduced in the previous section, co-player shaping becomes a conventional expected return maximization problem. Applying the policy gradient theorem (Sutton et al., 1999) to Eq. 2, we arrive at COALA-PG (co-agent learning-aware policy gradients, c.f. Theorem 3.1): a policy-gradient method compatible with shaping other reinforcement learners that base their own policy updates on minibatches of experienced trajectories.

**Theorem 3.1.** *Take the expected shaping return* $\bar{J}(\phi^i) = \mathbb{E}_{\bar{P}^{\phi^i}} \left[ \frac{1}{B} \sum_{b=1}^B \sum_{l=0}^{MT} R_l^{i,b} \right]$, *with* $\bar{P}^{\phi^i}$ *the distribution induced by the environment dynamics* $\bar{P}_t$, *initial state distribution* $\bar{P}_i$ *and policy* $\phi^i$.

**Naive agent:** Policy update after every inner-episode batch     **COALA-PG agent:** Policy update after every meta-episode

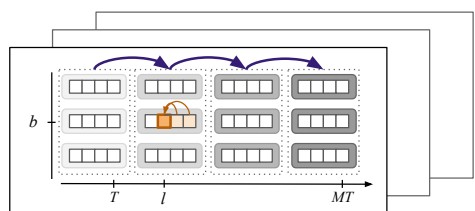
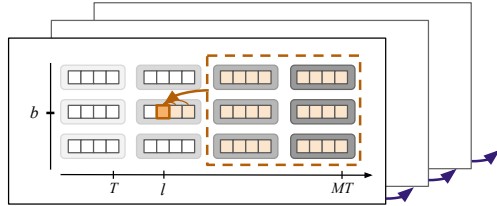

**Figure 2: Policy update and credit assignment of naive and meta agents.** For **credit assignment** of action $a_l^{i,b}$, a naive agent (left) takes only intra-episode context into account. A COALA agent (right) takes inter-episode context across the batch dimension into account. For **policy updates**, a naive agent aggregates policy gradients over the inner-batch dimension (dashed blocks) and updates their policy between episode boundaries. In contrast, a COALA agent updates their policy at a lower frequency along the meta-episode dimension.

*Then the policy gradient of this expected return is equal to*

$$\nabla_{\phi^i} \bar{J}(\phi^i) = \mathbb{E}_{\bar{P}^{\phi^i}} \left[ \sum_{b=1}^{B} \sum_{l=1}^{MT} \nabla_\phi \log \pi^i(a_l^{i,b} \mid h_l^{i,b}) \left( \frac{1}{B} \sum_{l'=l}^{m_l T} r_{l'}^{i,b} + \frac{1}{B} \sum_{b'=1}^{B} \sum_{l'=m_l T+1}^{MT} r_{l'}^{i,b'} \right) \right]. \quad (3)$$

We provide a proof in Appendix D. There are three important differences between COALA-PG and naively applying policy gradient methods to individual trajectories in a batch. (i) Each gradient term for an individual action $a_l^{i,b}$ takes into account the future inner episode returns averaged over the whole minibatch, instead of the future return along trajectory $b$ (see Fig. 2). This allows taking into account the influence of this action on the parameter update of the naive learner, which influences all trajectories in the minibatch after that update. (ii) Instead of averaging the policy gradients for each trajectory in the batch, COALA-PG accumulates (sums) them. This is important, as otherwise the learning signal would vanish in the limit of large minibatches. Intuitively, when a naive learner uses a large minibatch for its updates, the effect of a single action on the naive learner's update is small ($O(\frac{1}{B})$), and this must be compensated by summing all such small effects. (iii) To ensure a correct balance between the return from the current inner episode $m_l$ and the return from future inner episodes, COALA-PG rescales the current episode return by $\frac{1}{B}$. Figure 12 in App. H shows empirically that COALA-PG correctly balances the policy gradient terms arising from the current inner episode return versus the future inner episode returns, whereas M-FOS (Lu et al., 2022) and a naive policy gradient that ignores the other parallel trajectories over-emphasize the current inner episode return, causing them to loose the co-player shaping learning signals. We will later show experimentally in Section 5 that correct treatment of minibatches critically affects reinforcement learning performance.

The expectation appearing in the policy gradient expression must be estimated. To reduce gradient estimation variance, we resort to standard practices, including generalized advantage estimation (Schulman et al., 2016) and sampling a meta-batch of $\bar{B}$ batched trajectories from $\mathbb{E}_{\bar{P}^{\phi^i}}$ (c.f. Appendix A).

**Relationship to prior shaping methods.** We now contrast our policy gradient algorithm to two closely related methods, M-FOS (Lu et al., 2022) and Shaper (Khan et al., 2024). Like COALA-PG, M-FOS is a model-free meta reinforcement learning method. Unlike the approached followed here, though, it aims to solve the bilevel co-player shaping problem of Eq. 1, treating meta- and inner-policy networks separately. Moreover, the M-FOS parameter update is not derived as the policy gradient on the batched co-player shaping POMDP introduced above, and current-episode returns are overemphasized compared to future-episode returns (see Appendix G). This leads to a biased parameter update, which results in learning inefficiencies. We comment on other existing bilevel shaping methods in Appendix F.

Khan et al. (2024) adopt a single-level sequence policy for their Shaper algorithm, as we do here, but then resort to black-box evolution strategies (Rechenberg & Eigen, 1973) to learn the policy. Obtaining an efficient meta reinforcement learning algorithm from a POMDP applicable to such single-level policies is thus our key distinguishing contribution. The unbiased policy gradient property of our learning rule translates in practice onto learning speed and stability gains, as we will see in the experiments reported in Section 5.

## 4  WHY IS LEARNING AWARENESS BENEFICIAL ON GENERAL-SUM GAMES?

We have established that co-player shaping can be cast as a single-agent reward maximization problem whenever there is a single learning-aware player amongst a group of learners that are otherwise naive. This allowed us to derive a policy gradient shaping method. However, such an asymmetric setup cannot in general be taken for granted. In our experimental analyses, we therefore consider the more realistic scenario where equally-capable, learning-aware agents try to shape each other.

As reviewed in Section 2.2, prior work has shown that learning-awareness can result in better outcomes in general-sum games, but the origin and conditions for the occurrence of this phenomenon are not yet well understood. Here, we shed light on this question by analyzing the interactions of agents with varying degrees of learning-awareness in an analytically tractable matrix game setting. This leads us to uncover a novel explanation for the emergence of cooperation in general-sum games.

### 4.1  THE ITERATED PRISONER'S DILEMMA

We focus on the infinitely iterated prisoner's dilemma (IPD), the quintessential model for understanding the challenges of cooperation among self-interested agents (Rapoport, 1974; Axelrod & Hamilton, 1981). The game goes on for an indefinite number of rounds, where for each round of play two players ($i = 1, 2$) meet and choose between two actions, cooperate or defect, $a_t^i \in \{c, d\}$. The rewards collected as a function of the actions of both agents are shown in Table 1. These four rewards are set so as to create

**Table 1:** Single-round IPD rewards $(r^1, r^2)$.

|       | **c**     | **d**     |
|-------|-----------|-----------|
| **c** | (1,1)     | (-1,2)    |
| **d** | (2,-1)    | (0,0)     |

a social dilemma. When the agents meet only once, mutual defection is the only Nash equilibrium; self-interested agents thus end up obtaining low reward. In the infinitely iterated variant of the game, there exist Nash equilibria involving cooperative behavior, but these are notoriously hard to converge to through self-interested reward maximization.

We model each agent through a tabular policy $\pi^i(a_t^i \mid x_t^i; \phi^i)$ that depends only on the previous action of both agents, $x_t^i = (a_{t-1}^1, a_{t-1}^2)$. Their behavior is thus fully specified by five parameters, which determine the probability of cooperating in response to the four possible previous action combinations together with the initial cooperation probability. For this game, the discounted expected return $J^i(\phi^1, \phi^2)$ can be calculated analytically. We exploit this property and optimize policies by performing exact gradient ascent on the expected return (c.f. Appendix C for details).

### 4.2  EXPLAINING COOPERATION THROUGH LEARNING AWARENESS

Based on the experimental results reported in Fig. 3, we now identify three key findings that establish how learning awareness enables cooperation to be reached in the iterated prisoner's dilemma:

**Finding 1: Learning-aware agents extort naive learners.** We first pit naive against learning-aware agents. We find that the latter develop extortion policies which force naive learners onto unfair cooperation, similar to the zero-determinant extortion strategies discovered by Press & Dyson (2012) (c.f. Appendix C.1). Even when a learning-aware agent is initialized at pure defection, maximizing the shaping objective of Eq. 2 lets it escape mutual defection (see Fig. 3A).

**Finding 2: Extortion turns into cooperation when two learning-aware players face each other.** After developing extortion policies against naive learners (grey shaded area in Fig. 3B), we then let two learning-aware agents (C1 and C2 in Fig. 3B) play against each other after. We see that optimizing the co-player shaping objective turns extortion policies into cooperative policies. Intuitively, under independent learning, an extortion policy shapes the co-player to cooperate more. We remark that the same occurs if learning-aware agents play against themselves (self-play; data not shown). This analysis explains the success of the annealing procedure employed by Lu et al. (2022), according to which naive co-players transition to self-play throughout training.

**Finding 3: Cooperation emerges within groups of naive and learning-aware agents.** Findings 1. and 2. motivate studying learning in a group containing both naive and learning-aware agents, with every agent in the group trained against each other. This mixed group setting yields a sum of two distinct shaping objectives, which depend on whether the agent being shaped is learning-aware or naive. The gradients resulting from playing against naive learners pull against mutual defection and towards extortion, while those resulting from playing against other learning-aware agents push away from extortion towards cooperation. Balancing these competing forces leads to robust cooperation, see Fig. 3C (left). Intriguingly, mutual unconditional defection is no longer a Nash equilibrium

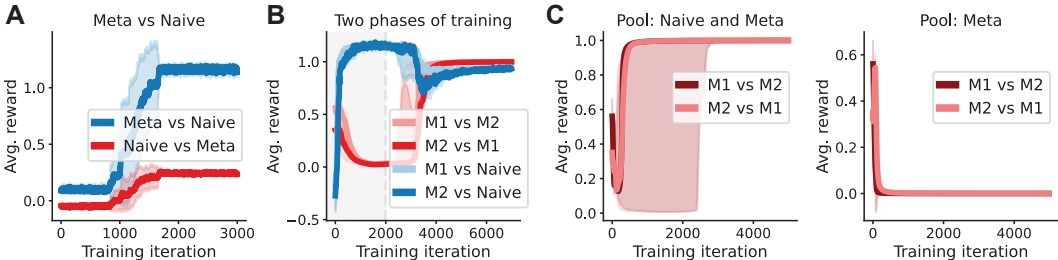

**Figure 3:** (A) Learning-aware agents learn to extort naive learners, even when initialized with pure defection strategy. (B) An extortion policy developed against naive agents (shaded area period) turns into a cooperative one when playing against another learning-aware agent (M1 & M2). (C) Cooperation emerges within mixed training pools of naive and learning-aware agents, but not in pools of learning-aware agents only. The shaded regions represent the interquartile range (25th to 75th quantiles) across 32 random seeds

in this mixed group setting, and agents initialized with unconditional defection policies learn to cooperate (see Appendix C.1). By contrast, a pure group of learning-aware agents cannot escape mutual defection, see Fig. 3C (right). This can be explained by the fact that the agents can no longer observe others learn, and must deal again with a non-stationary problem. The resulting gradients do not therefore contain information on the effects of unconditional defection on the future strategies of co-players, or that policies in the vein of tit-for-tat can shape co-players towards more cooperation.

Our analysis thus reveals a surprising path to cooperation through heterogeneity. The presence of short-sighted agents that greedily maximize immediate rewards turns out to be essential for full cooperation to be established among far-sighted, learning-aware agents.

### 4.3 EXPLAINING WHEN AND HOW COOPERATION ARISES WITH THE LOLA ALGORITHM

We next analyze the seminal Learning with Opponent-Learning Awareness (LOLA; Foerster et al., 2018a) algorithm. Briefly, LOLA assumes that co-players update their parameters with $M$ naive gradient steps, and estimates the total gradient through a look-ahead update:

$$\nabla_{\phi^i}^{\text{LOLA}} = \frac{\mathrm{d}}{\mathrm{d}\phi^i} \left[ J^i \left( \phi^i, \phi^{-i} + \sum_{q=1}^{M} \Delta_q \phi^{-i} \right) \quad \text{s.t. } \Delta_q \phi^{-i} = \alpha \frac{\partial}{\partial \phi^{-i}} J^i \left( \phi^i, \phi^{-i} + \sum_{q'=1}^{q-1} \Delta_{q'} \phi^{-i} \right) \right] \quad (4)$$

with $\frac{\mathrm{d}}{\mathrm{d}\phi^i}$ the total derivative taking into account the effect of $\phi^i$ on the parameter updates $\Delta_q \phi^{-i}$, and $\frac{\partial}{\partial \phi^{-i}}$ the partial derivative. Note that Eq. 4 considers the LOLA-DICE update (Foerster et al., 2018b), an improved version of LOLA. In Appendix E, we show that Eq. 4 can be derived as a special case of COALA-PG. Note that LOLA-DICE estimates the policy gradient in Eq. 4 by explicitly backpropagating through the co-player's parameter update using higher-order derivatives, whereas COALA-PG leads to a novel higher-order-derivative-free estimator of Eq. 4 (see Appendix E).

Above, we showed that the two main ingredients for learning to cooperate under selfish objectives are (i) observe that one's actions influence the future behavior of others, providing shaping gradients pulling away from defection towards extortion, and (ii), also play against other extortion agents immune to being shaped on the fast timescale, providing gradients pulling away from extortion towards cooperation. We then showed that both ingredients can be combined by training agents in a heterogeneous group containing both naive and learning-aware agents.

We can explain the emergent cooperation in LOLA by observing that LOLA also combines both ingredients, albeit differently from the heterogeneous group setting. The look-ahead rule (Eq. 4) computes gradients that shape naive learners performing $M$ naive gradient steps. Unique to LOLA however, these simulated naive learners are initialized with the parameters $\phi^{-i}$ of other LOLA agents. If the number of look-ahead steps is small, the updated naive learner parameters stay close to $\phi^{-i}$, mimicking playing against other extortion agents. This then results in emergent cooperation.

Fig. 4A confirms that LOLA-DICE with ground-truth gradients and with few look-ahead steps leads to cooperation on the iterated prisoner's dilemma. However, as the number of look-ahead steps increases, the naive learner starts moving too far away from its $\phi^{-i}$ initialization, removing the second ingredient, thus leading to defection. In Fig. 4, we take the policy resulting from LOLA-training with many look-ahead steps, and train a new randomly initialized naive learner against this

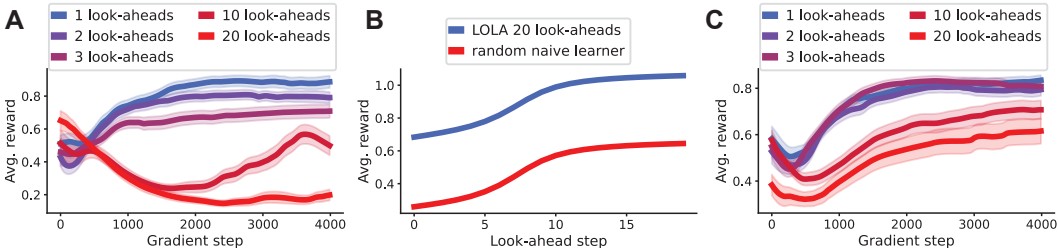

**Figure 4:** (A) Performance of two agents trained by LOLA-DICE on the iterated prisoner's dilemma with analytical gradients for various look-ahead steps (only the performance of the first agent is shown). (B) Performance of a randomly initialized naive learner trained against the fixed LOLA 20 look-aheads policy taken from the end of training of (A). (C) Same setting as (A), but with the naive gradient $\lambda \frac{\partial}{\partial \phi^{-i}} J^i(\phi, \phi^{-i})$ added to the LOLA-DICE update, with $\lambda$ a hyperparameter (c.f. Appendix C). Shaded regions indicate standard error computed over 64 seeds.

fixed LOLA policy. The results show that the LOLA policy extorts the naive learner into unfair cooperation, confirming that with many look-ahead steps, only a shaping incentive is present in the LOLA update, resulting in extortion policies. Hence, the low reward in Fig. 4A for LOLA agents with many look-ahead steps does not result from unconditional defection, but instead from both LOLA policies trying to extort the other one. Finally, we can improve the performance of LOLA with many look-ahead steps by explicitly introducing ingredient 2 by adding the partial gradient $\frac{\partial}{\partial \phi^{-i}} J^i(\phi, \phi^{-i})$ to Eq. 4, see Fig. 4C.

# 5 EXPERIMENTAL ANALYSIS OF POLICY GRADIENT IMPLEMENTATIONS

The results presented in the previous sections were obtained by performing gradient ascent on analytical expected returns. This assumes knowledge of co-player parameters, and it is only possible on a restricted number of games which admit closed-form value functions. We now move to the general reinforcement learning setting, aiming at understanding (i) when meta-agents succeed in exploiting naive agents, and (ii) when cooperation is achieved among meta-agents.

## 5.1 AGENTS TRAINED WITH COALA-PG MASTER THE ITERATED PRISONER'S DILEMMA

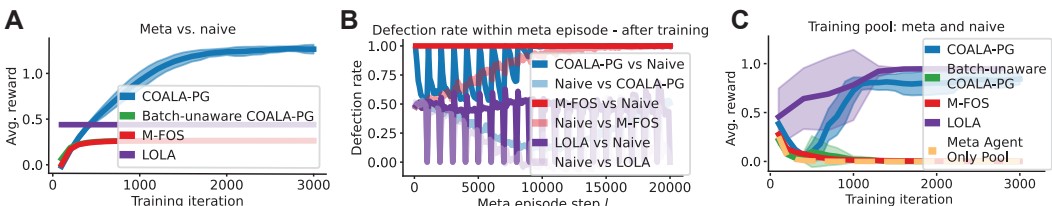

**Figure 5: Agents trained by COALA-PG play iterated prisoner's dilemma.** (A): When trained against naive agents only, COALA-PG-trained agents extort the latter and reach considerably higher reward than other baseline agents. The stars (⋆) indicate overlapping curves of the corresponding color at that point (B): When analyzing the behavior of the agents within one meta-episode, we observe COALA-PG-trained agents shaping naive co-players, leading to low defection rate in the beginning, which is then exploited towards the end. M-FOS on the other hand defects from the beginning, achieving lower reward, thus failing to properly optimize the shaping problem. Batch-unaware COALA-PG performs identically to M-FOS and is therefore omitted. (C): Average performance of meta agents playing against other meta agents, when training a group of meta agents against a mixture of naive and other meta agents. Such agents trained with COALA-PG cooperate when playing against each other, but fail to do so when trained with baseline methods. When removing naive agents from the pool, meta agents also fail to cooperate, as predicted in Section 3. Shaded regions indicate standard deviation computed over 5 seeds.

We train a long-context sequence policy $\pi^i(a_l^{i,b} \mid h_l^{i,b}; \phi^i)$ with the COALA-PG rule to play the (finite) iterated prisoner's dilemma, see Appendix B. We choose a Hawk recurrent neural network as the policy backbone (De et al., 2024). Hawk models achieve transformer-level performance at scale, but with time and memory costs that grow only linearly with sequence length. This allows processing efficiently the long history context $h_l^{i,b}$, which contains all actions played by the agents

across episodes. Based on the results of the preceding section, we consider a mixed group setting, pitting `COALA-PG`-trained agents against naive learners as well as other equally capable learning-aware agents. Naive learners are equipped with the same policy architecture as the agents trained by `COALA-PG`, but their context is limited to the current inner game history $x_t^{i,b}$.

In Fig. 5, we see that `COALA-PG` reproduces the analytical game findings reported in the previous section: learning-aware agents cooperate with other learning-aware agents, and extort naive learners. Importantly, the identity of each agent is not revealed to a learning-aware agent, which must therefore infer in-context the strategy used by the player it faces. Likewise, we find that the LOLA-DICE (Eq. 4) estimator behaves as in the analytical game. This result complements previous experiments with LOLA on tabular policies (Foerster et al., 2018a;b), suggesting that there is a broad class of efficient learning-aware reinforcement learning rules that can reach cooperation with more complex context-dependent sequence models. We note that LOLA achieves this by explicitly differentiating through co-player updates, which requires access to their parameters and gives rise to higher-order derivatives. Our rule lifts these requirements, while maintaining learning efficiency.

By contrast, the whole group falls into defection when training the exact same sequence model with the M-FOS rule, which weighs disproportionately future vs. current episode returns. We note that the experiments reported by Lu et al. (2022) were performed with a tabular policy and analytical inner game returns, for which cooperation could be achieved with the M-FOS rule. This shows how crucial the unbiased policy gradient property of `COALA-PG` is for co-player shaping by meta reinforcement learning to succeed in practice. The same failure to beat defection occurs when using a naive policy gradient ablation which does not take co-player batching into account. We refer to Appendix G for expressions for this baseline as well as the M-FOS rule. When training against M-FOS agents, our `COALA-PG` agents successfully shape M-FOS agents into cooperative behavior (c.f. Appendix H.3).

## 5.2 AGENTS TRAINED WITH `COALA-PG` COOPERATE ON A SEQUENTIAL SOCIAL DILEMMA

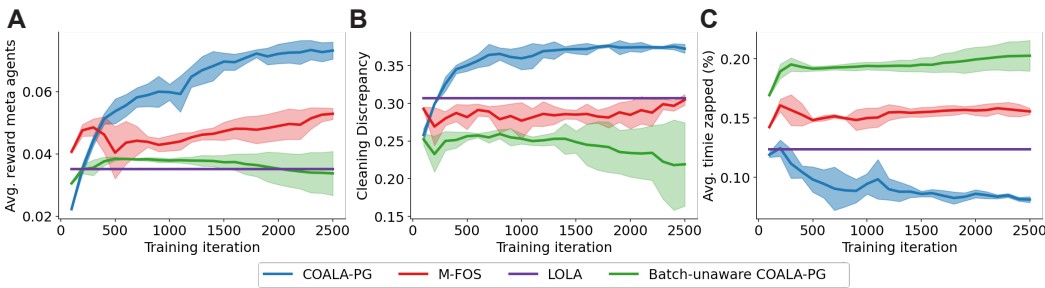

**Figure 6: Agents trained by `COALA-PG` against naive agents only successfully shape them in `CleanUp-lite`.** (A) `COALA-PG`-trained agents better shape naive opponents compared to baselines, obtaining higher return. (B and C) Analyzing behavior within a single meta-episode after training reveals that COALA outperforms baselines and shapes naive agents, (i) exhibiting a lower cleaning discrepancy (absolute difference in average cleaning time between the two agents), and (ii) being less often zapped. Shaded regions indicate standard deviation computed over 5 seeds.

Finally, we consider `CleanUp-lite`, a simplified two-player version of the `CleanUp` game, which is part of the Melting Pot suite of multi-agent environments (Agapiou et al., 2023). We briefly describe the game here, and provide additional details on Appendix B. On a high level, `CleanUp-lite` is a two-player game that models the social dilemma known as the tragedy of the commons (Hardin, 1968). A player receives rewards by picking up apples. Apples are spontaneously generated in an orchard, but the rate of generation is inversely proportional to the pollution level of a nearby river. Agents can spend time cleaning the river to reduce the pollution level, and thus increase the apple generation rate. In a single-player game, an agent would balance out cleaning and harvesting to maximize the return. In a multi-player setting however, this gives room for a "freerider" who never cleans and always harvests, letting the opponent clean instead. At any time point, agents can "zap" the opponent, which would result in the opponent being frozen for a number of time steps, unable to harvest or clean. In contrast to matrix games, this game is a sequential social dilemma (Leibo et al., 2017), where cooperation involves orchestrating multiple actions.

As in the previous section, we model agent behavior through Hawk sequence policies, and compare `COALA-PG` to the same baseline methods as before. Naive agents here would learn too slowly if initialized from scratch, and are therefore handled differently, see Appendix B.2.2. In Figs. 6 and 7, we see that agents trained by COALA-PG reach significantly higher returns than previous model-free baselines, establishing a mutual cooperation protocol with other learning-aware agents, while exploiting naive ones. We further describe below the qualitative behavior found in the simulations.

**Exploitation of naive agents.** `COALA-PG`-trained agents shape the behavior of naive ones to their advantage (c.f. Figure 6). Our behavioral analysis reveals two salient features. First, `COALA-PG` successfully shapes naive opponents to zap less often throughout the meta-episode. Less overall zapping means that agents can harvest more apples while the pollution level is low, thus increasing the overall reward. Second, `COALA-PG` successfully shapes naive co-players to clean significantly more often compared to the `COALA-PG` agent, resulting in a lower average pollution level and a higher average apple level (c.f. Figure 13 in App. H). Interestingly, the naive learners benefit from the shaping from `COALA-PG` agents, reaching a higher average reward compared to playing against other baselines (c.f. Figure 13).

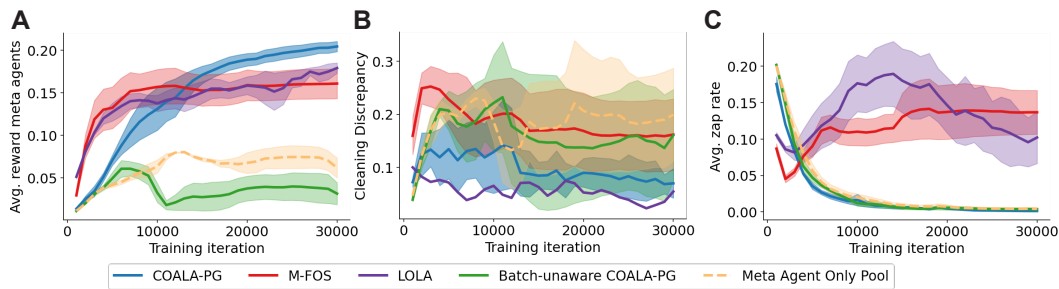

**Figure 7: Agents trained with `COALA-PG` against a mixture of naive and other meta agents learn to cooperate in `CleanUp-lite`.** (A) `COALA-PG`-trained agents obtain higher average reward than baseline agents when playing against each other. (B and C): `COALA-PG` leads to a more fair division of cleaning efforts and lower zapping rates. Shaded regions indicate standard deviation computed over 5 seeds.

**Learning-aware agents cooperate.** We see similar trends when introducing other `COALA-PG`-trained agents in the game, see Fig. 7. Essentially, `COALA-PG` allows for higher apple production because of lower pollution, and lower zapping rate. We see that over training time the zapping rate goes down, and `COALA-PG` agents have a fairer division of cleaning time compared to baselines. Interestingly, the zapping rates averaged over the meta-episode are lower than in the pure shaping setting (i.e., with naive co-players only), indicating that learning-aware agents mutually shape each other to zap less.

## 6 CONCLUSION

We have shown that learning awareness allows reaching high returns in challenging social dilemmas, designed to make independent learning difficult. We identified two key conditions for this to occur. First, we found it necessary to take into account the stochastic minibatched nature of the updates used by other agents. This is one distinguishing aspect of the COALA-PG learning rule proposed here, which translates into a significant performance advantage over prior methods. Second, learning-aware agents had to be embedded in a heterogeneous group containing non-learning-aware agents.

An important component of our result is the ability to leverage modern and scalable sequence models. Modern sequence models have scaled favorably and in a predictable manner, most notably in autoregressive language modeling (Kaplan et al., 2020), and our results suggest important gains could be made applying similar approaches to multi-agent learning. Our method shares key aspects with the current scalable machine learning approach: unbiased stochastic gradients, sequence model architectures that are amenable to gradient-based learning, and in-context learning/inference. Moreover, we focused on the setting of independent agent learning, which scales well in parallel by design. We thus see it as an exciting question to investigate the approach pursued here at larger scale and in a wider range of environments. The resulting self-organized behavior may display unique social properties that are absent from single-agent machine learning paradigms, and which may open new avenues towards artificial intelligence (Duéñez-Guzmán et al., 2023).

ACKNOWLEDGEMENTS

We would like to thank Maximilian Schlegel, Yanick Schimpf, Rif A. Saurous, Joel Leibo, Alexander Sasha Vezhnevets, Aaron Courville, Juan Duque, Milad Aghajohari, Razvan Ciuca, Gauthier Gidel, James Evans and the Google Paradigms of Intelligence team for feedback and enlightening discussions. GL and BR acknowledge support from the CIFAR chair program. EE acknowledges support from a Vanier scholarship from the government of Canada.

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

## A  A2C AND PPO IMPLEMENTATIONS OF COALA-PG

We use both Advantage Actor-Critic (A2C) (Mnih et al., 2016) and Proximal Policy Optimization (PPO) Schulman et al. (2017) for our COALA policy gradient estimate. We detail here how to merge these methods with our COALA-PG method.

### A.1  REINFORCE ESTIMATOR

For the reader's convenience, we display the COALA policy gradient below. We remind for reference, that $m_l$ the inner episode index corresponding to the meta episode time step $l$.

$$\nabla_{\phi^i} \bar{J}(\phi^i) = \mathbb{E}_{\bar{P}\phi^i} \left[ \sum_{b=1}^{B} \sum_{l=1}^{MT} \nabla_{\phi^i} \log \pi^i(a_l^{i,b} \mid h_l^{i,b}) \left( \frac{1}{B} \sum_{k=l}^{Tm_l} R_k^{i,b} + \frac{1}{B} \sum_{b'=1}^{B} \sum_{k=Tm_l+1}^{MT} R_k^{i,b'} \right) \right]. \tag{5}$$

The batch-unaware COALA policy gradient, which we use as a baseline method for shaping naive learners, is given by

$$\nabla_{\phi^i} \bar{J}(\phi^i) = \mathbb{E}_{\bar{P}\phi^i} \left[ \frac{1}{B} \sum_{b=1}^{B} \sum_{l=1}^{MT} \nabla_{\phi^i} \log \pi^i(a_l^{i,b} \mid h_l^{i,b}) \left( \sum_{k=l}^{Tm_l} R_k^{i,b} + \sum_{k=Tm_l+1}^{MT} R_k^{i,b} \right) \right]. \tag{6}$$

Note that when we play against other meta agents instead of naive learners, all parallel POMDP trajectories in the batch are independent, and hence we can correctly use the batch-unaware COALA policy gradient for this setting.

Finally, the M-FOS policy gradient (c.f. Appendix G) is given by

$$\nabla_{\phi^i} \bar{J}(\phi^i) = \mathbb{E}_{\bar{P}\phi^i} \left[ \sum_{b=1}^{B} \sum_{l=1}^{MT} \nabla_{\phi^i} \log \pi^i(a_l^{i,b} \mid h_l^{i,b}) \left( \sum_{k=l}^{Tm_l} R_k^{i,b} + \frac{1}{B} \sum_{b'=1}^{B} \sum_{k=Tm_l+1}^{MT} R_k^{i,b'} \right) \right]. \tag{7}$$

The difference between M-FOS and COALA-PG is the $\frac{1}{B}$ scaling factor for the current inner episode return. This scaling factor is crucial for a correct balance between gradient contributions arising from the current inner episode, and future inner episodes. Without this scaling factor, the contributions from future inner episodes required for learning to shape the co-players vanish for large inner batch sizes.

We can construct REINFORCE estimators by sampling directly from the above expectations. However, this leads to policy gradients with prohibitively large variance. Hence, in the following sections we will derive improved advantage estimators to reduce the variance of the policy gradient estimates.

### A.2  VALUE FUNCTION ESTIMATION

One of the easiest ways to use value functions for reducing the variance in the policy gradient estimator, is to subtract a baseline from the return estimator. In the COALA-PG equation 5, the straightforward value function to learn is

$$V(h_l^{i,b}) = \mathbb{E}_{\bar{P}\phi^i(\cdot|h_l^{i,b})} \left[ \left( \frac{1}{B} \sum_{k=l}^{Tm_l} R_k^{i,b} + \frac{1}{B} \sum_{b'=1}^{B} \sum_{k=Tm_l+1}^{MT} R_k^{i,b'} \right) \right]. \tag{8}$$

As the environment is reset after each inner episode, the second term can be simplified by merging expectations over the different parallel trajectories:

$$\mathbb{E}_{\bar{P}\phi^i(\cdot|h_l^{i,b})} \left[ \left( \sum_{k=l}^{Tm_l} \frac{1}{B} R_k^{i,b} + \sum_{k=Tm_l+1}^{MT} R_k^{i,b'} \right) \right]. \tag{9}$$

Which has an additional $\frac{1}{B}$ factor on the left term compared to a conventional value function that would need to be learnt when playing e.g. against another meta agent equation 6. This is undesirable for several reasons, one of which being that as $B$ increases, the value target becomes increasingly insensitive to the inner episode return, which makes learning difficult. Another reason is that the target magnitude between when playing against a naive agent or a meta agent can significantly differ. Finally, for simplicity reasons, we want a value function that we can use both when playing against naive learners, as well as other meta agents.

We can solve these issues by instead learning the value function for the batch-unaware returns, and introducing some specialized reweighing when playing against naive learners, which we will see later.

$$\hat{V}(h_l^{i,b}) = \mathbb{E}_{\bar{P}^{\phi^i}(\cdot|h_l^{i,b})}\left[\left(\sum_{k=l}^{Tm_l} R_k^{i,b} + \sum_{k=Tm_l+1}^{MT} R_k^{i,b}\right)\right]. \tag{10}$$

As such, the same value function can be used for both when playing against a naive or a meta agent. In practice, we trade off variance and bias for learning the value function by using TD($\lambda$) targets. Algorithm 1 shows how to compute such targets with a general algorithm, which we later can also repurpose for Generalized Advantage Estimation (Schulman et al., 2016) and M-FOS value functions. For computing the TD($\lambda$) targets for learning our value functions, we use `normalize_current_episode=False` and `average_future_episodes=False` when the given trajectory batch originates from playing against another meta agent.

---

**Algorithm 1** Batch Lambda Returns

---

**Input:** $r_t$, $discount$, $v_t$, $\lambda$, $average\_future\_episodes$, $normalize\_current\_episode$, $inner\_episode\_length$
**Output:** returns
$seq\_len \leftarrow r_t.shape[1]$
$batch\_size \leftarrow r_t.shape[0]$
**if** $normalize\_current\_episode$ **then**
  $\quad normalization \leftarrow batch\_size$
**else**
  $\quad normalization \leftarrow 1$
$episode\_end \leftarrow (\textbf{range}(seq\_len) \mod inner\_episode\_length) == (inner\_episode\_length - 1)$
$acc \leftarrow v_t[:, -1]$
$global\_acc \leftarrow mean(v_t[:, -1])$
**for** $t = seq\_len - 1$ **to** $0$ **do**
  $\quad$ **if** $average\_future\_episodes$ **and** $episode\_end[t]$ **then**
  $\qquad acc \leftarrow global\_acc$

  $\quad acc \leftarrow r_t[:, t]/normalization + discount \times ((1 - \lambda) \times v_t[:, t] + \lambda \times acc)$
  $\quad global\_acc \leftarrow mean(r_t[:, t] + discount \times ((1 - \lambda) \times v_t[:, t] + \lambda \times global\_acc))$
  $\quad returns[:, t] \leftarrow acc$

**return** $returns$

---

### A.3 GENERALIZED ADVANTAGE ESTIMATION

We now see how the above value estimation can be used to update the policy following `COALA-PG`. Ultimately we want an unbiased estimate of the advantage function, as this allows the usage of algorithms like PPO or A2C.

The advantage of a state $h_l^{i,b}$ and action $a_l^{i,b}$ against a naive agent is

$$A(h_l^{i,b}, a_l^{i,b}) = \mathbb{E}_{\bar{P}^{\phi^i}(\cdot|h_l^{i,b}, a_l^{i,b})} \left[ \frac{1}{B} \sum_{k=l}^{Tm_l} R_k^{i,b} + \frac{1}{B} \sum_{b'=1}^{B} \sum_{k=Tm_l+1}^{MT} R_k^{i,b'} \right] \tag{11}$$

$$- \mathbb{E}_{\bar{P}^{\phi^i}(\cdot|h_l^{i,b})} \left[ \frac{1}{B} \sum_{k=l}^{Tm_l} R_k^{i,b} + \frac{1}{B} \sum_{b'=1}^{B} \sum_{k=Tm_l+1}^{MT} R_k^{i,b'} \right]. \tag{12}$$

We can reformulate the expression using $\hat{V}$ as follows:

$$A(h_l^{i,b}, a_l^{i,b}) = \mathbb{E}_{\bar{P}^{\phi^i}(\cdot|h_l^{i,b}, a_l^{i,b})} \left[ \frac{1}{B} \sum_{k=l}^{Tm_l} R_k^{i,b} + \frac{1}{B} \sum_{b'=1}^{B} \sum_{k=Tm_l+1}^{MT} R_k^{i,b'} \right] \tag{13}$$

$$- \frac{1}{B} \hat{V}(h_l^{i,b}) - \frac{1}{B} \mathbb{E}_{\bar{P}^{\phi^i}(\cdot|h_l^{i,b})} \left[ \sum_{b' \neq b} \hat{V}(h_{Tm_l+1}^{i,b'}) \right]. \tag{14}$$

A simple advantage estimator would be the Monte-Carlo estimate of the above. However, we can trade-off variance with bias by using the Generalized Advantage Estimator (Schulman et al., 2016). Using similar logic as for the equation above, we can compute the COALA version of the GAE by reusing the `batched_lambda_returns` algorithm (c.f. Algorithm 1 as follows:

- Instead of the rewards of the trajectory, we provide the TD errors $\delta_t = r_t + \gamma \hat{V}_{t+1} - \hat{V}_t$ as input for `r_t`.

- We provide $\gamma \lambda$ as input for `discount`

- We provide $1.0$ as input for $\lambda$

- We put `average_future_episodes` and `normalize_current_episode` both on `True`.

For computing the GAE for the batch-unaware COALA-PG baseline, we follow the same approach except putting `average_future_episodes` and `normalize_current_episode` both on `False`. For computing the GAE for the M-FOS baseline (c.f. G), we follow the same approach except putting `average_future_episodes` on `True` and `normalize_current_episode` on `False`.

## A.4 A2C AND PPO IMPLEMENTATIONS

We can now use the above advantage estimates directly into A2C and PPO implementations. Below, we list a few tweaks of classical reinforcement learning tricks that we used in our implementation.

- Advantage normalization: as is common with PPO implementations, we investigate the use of advantage normalization. Given a batched trajectory of advantage estimation over which the policy should be updated, the trick consists in centering the advantage estimates over the batched trajectory. Empirically, we found out that when playing against a mixture of naive and meta learners, it was beneficial to apply the centering separately for the 2 types of meta-trajectories (playing against naive learners or playing against other meta agents).

- Reward rescaling: as another way to prevent issues stemming from large value target, we investigate simply rescaling the reward of an environment when appropriate. Effectively, the reward is rescaled for the value and policy gradient computation, but all metrics are reported by reverting the scaling, i.e. reported in the original reward scale.

# B EXPERIMENTAL DETAILS

## B.1 ENVIRONMENTS

### B.1.1 ITERATED PRISONER'S DILEMMA (IPD)

We model the IPD environment as follows:

- **State:** The environment has 5 states, that we label by $s_0, (c, c), (c, d), (d, c), (d, d)$.
- **Action:** Each agent has 2 possible actions: cooperate ($c$) and defect ($d$).
- **Dynamics:** Based on the action taken by each agent in the previous time step, the state of the environment is set to the states $(a_1, a_2)$ where $a_1, a_2$ are respectively the previous action of the first and second player in the environment. The assignment of who is first and second is made arbitrarily and fixed.
- **Initial state:** The initial state is always set to $s_0$.
- **Observation:** The agents observe directly the state, modulo a permutation of the tuple to ensure a symmetry of observation. The 5 possible observations are then encoded as one-hot encoding.
- **Reward:** At every timestep, each agents receive a reward following the reward matrix in Table 1

### B.1.2 CLEANUP-LITE

`CleanUp-lite` is a simplified two-player version of the `CleanUp` game, which is part of the Melting Pot suite of multi-agent environments (Agapiou et al., 2023). It is modelled as follows:

- **State:** The world is a 2D grid of size 54. The right column is the river, and the left one the orchard. Cells in the river column can be occupied by dirt. Cells in the orchard column can be occupied by an apple. The world state also contains the position of each agent, and their respective zapped state
- **Action:** There exists 6 actions: {move right, move left, move up, moved down, zap, do nothing}.
- **Dynamics:** the environment evolves at every timestep in the following order:
  1. When there is at least one cell in the river column that is not occupied by dirt, a new patch of dirt is spawned with probability $p_{\text{pollution}} = 0.35$, and placed randomly on one of the free cells in the river column.
  2. When there is at least one cell in the orchard column that is not occupied by dirt, a new apple is spawned with probability $p_{\text{apple}} = 1 - \min(1, P/P_{\text{threshold}})$, where $P_{\text{threshold}} = 3$ and $P$ the total number of dirt cells in the environment. The spawned apple is placed randomly on one of the free cells in the orchard column.
  3. When an agent that is not zapped visits a cell with an apple, it harvests the apple and gets a reward of 1. The apple is replaced by an empty cell
  4. When an agent that is not zapped visits a cell with a dirt patch, it cleans the dirt patch and replaces it by an empty cell.
  5. Finally, an agent zapping has a $p_{\text{zap}} = 0.9$ probability of successfully zapping the co-player, if the co-player is maximally 2 cells away from the agent. If the zapping is successful, the opponent is frozen for $t_{\text{zap}} = 5$ timesteps, during which it is frozen and cannot be further zapped.
  6. Agents can move around with the {move right, move left, move up, moved down} actions.
- **Initial state:** Agents are randomly placed on the grid, unzapped, there are no apples at initialization and 3 dirt patches randomly placed in the river column.
- **Observation:** the observation contains full information about the environment. Each agent sees the position of each agent encoded as flattened one-hot grid indicating the position in the grid, the full grid as a flattened grid with one-hot objects (apple, dirt, empty), and the state of all agents (zapped or non-zapped). The observation is symmetric.

- **Reward:** An agent that picks up an apple receives a reward of $r_{\text{apple}} = 1$ in that timestep.

## B.2 TRAINING DETAILS

Here, we describe the procedure that we use in our experiments to train meta agents in an arbitrary mixture of naive and other meta agents (who themselves are learning). A single parameter, $p_{\text{naive}}$, indicating the probability of encountering a naive agent, controls the heterogeneity of the pool that a meta agents trains against. If $p_{\text{naive}} = 1$, the meta agents are trained only against naive opponents, and thus the training corresponds to a pure shaping setting. If $p_{\text{naive}} = 0$, meta agents are only trained against other meta agents.

Given a set of meta agent parameters $\{\phi_i\}$, and a set of naive agent parameters $\{\psi_i\}$, a training iteration updates each parameters as follows.

### B.2.1 META AGENTS

The meta-agent parameters are updated simultaneously. For each parameter $\phi_i$, the following update is applied:

1. First, a meta batch of opponents is sampled. Each opponent is hierarchically sampled by first determining whether it is a naive opponent (with probability $p_{\text{naive}}$), and then sampling uniformly from $\{\phi_i\}$ or $\{\psi_i\}$ accordingly. The sampling is done with replacement, and disallowing sampling of oneself.

2. For each opponent, generate a batch of $B$ trajectories of length $TM$, where $M$ is the number of inner episode, and $T$ the episode length of the environment. Crucially, after every $T$ steps, the environment terminates and is reset, and, if the opponent is naive, the previous batch of length $T$ trajectories is used to update its parameter following a RL update rule of choice.

3. For each collected batched trajectories, the policy gradient of the meta agent parameter is computed following the COALA-PG update rule (or other baselines, c.f. G) if the opponent is naive, and the standard policy gradient otherwise otherwise (i.e. the batch and meta batch dimensions are flattened). Crucially, the done signals from the inner episodes are ignored. The gradient is then averaged, and the parameter updated.

### B.2.2 NAIVE AGENT

The naive agent parameters are used to initialize the naive opponents when training the meta agents, but the resulting trained parameters are discarded. The initialization may or may not be nonetheless updated during training. In a more challenging environment however, training from scratch until good performance is achieved in a single meta trajectory may require prohibitively many inner episodes. To avoid this, in some of our experiments, at each training iteration, we set each $\{\psi_i\}$ to be equal to one of the $\{\phi_i\}$. This ensures that naive agents are initialized as an already capable agent, and is possible due to our choice of common architecture between naive and meta agents (c.f. below). In that case, we say that the naive agents are **dynamic**. Otherwise, naive agent is always initailized at one of a predefined static set of parameters.

## B.3 ARCHITECTURE

We choose a Hawk recurrent neural network as the policy and value function backbone (De et al., 2024), for all methods, both for meta and naive agents. First, a linear layer projects the observation into an embedding space of dimension 32. Then, a single residual Hawk recurrent neural network with LRU width 32, MLP expanded width 32 and 2 heads follows. Finally, an RMS normalization layer is applied, after which 2 linear readouts, one for the value estimate, and the other for the policy logits, are applied.

All meta agent and naive agent parameters are initialized following the standard initialization scheme of Hawk. The last readout layers are however initialized to 0.

**Table 2:** Hyperparameter fixed for the IPD experiments

| IPD Hyperparameter | Pure Shaping | Mixed Pool |
|---|---|---|
| `training_iteration` | 3000 | 3000 |
| `meta_batch_size` | 128 | 128 |
| `batch_size (B)` | 16 | 16 |
| `num_inner_episode (M)` | 20 | 20 |
| `inner_episode_length (T)` | 10 | 10 |
| `p_naive` | 1. | 0.75 |
| `population_size (meta)` | 1 | 4 |
| `population_size (naive)` | 10 | 10 |
| `dynamic_naive_agents` | False | False |

### B.4 HYPERPARAMETERS FOR EACH EXPERIMENT

In all experiments, we first fix the environment hyperparameters. In order to find the suitable hyperparameter for each methods, we perform for each of them a sweep over reinforcement learning hyperparameters, and select the best hyperparameters over after averaging over 3 seeds. The final performance and metrics are then computed using 5 fresh seeds.

In all our experiments, naive agents update their parameters using the Advantage Actor Critic (A2C) algorithm, without value bootstrapping on the batch of length T trajectories. The hyperparameter for all experiments, can be found on Table 8.

**IPD, Figure 5**    We perform 2 experiments in the IPD environment, (i) the pure Shaping experiment with $p_{\text{naive}} = 1$ to investigate the shaping capabilities of meta agents, and (ii) the mixed pool setting with $p_{\text{naive}} = 0.75$, to investigate the collaboration capabilities of meta agents. For both experimental setting, we show the environment hyperparameters in Table 2. All meta agents are trained by PPO and Adam optimizer. For each method, we sweep hyperparameters over range specified in Table 3. Table 4 shows the resulting hyperparameters for all methods.

**Table 3:** The range of values swept over for hyperparameter search for each method for the IPD environment

| RL Hyperparameter | Range |
|---|---|
| `advantages_normalization` | $\{False, True\}$ |
| `value_discount` ($\gamma$) | $\{0.999, 1.0\}$ |
| `gae_lambda` ($\lambda_{\text{gae}}$) | $\{0.98, 1.0\}$ |
| `learning_rate` | $\{0.003, 0.001, 0.0003\}$ |

**Cleanup, Figure 6, 7**    Likewise, we have the pure shaping (Figure 6) and mixed pool (Figure 7) experiment in the Cleanup-lite environment. For both experimental setting, we show the environment hyperparameters in Table 5. All meta agents are trained by PPO and Adam optimizer for the pure shaping setting, while using A2C and SGD for the mixed pool setting. For each method, we sweep hyperparameters over range specified in Table 6. Table 7 shows the resulting hyperparameters for PPO for all methods.

**Table 4:** Hyperparameters used for the IPD Shaping and Mixed Pool experiments. Despite the search, the hyperparameter chosen for each method were identical

| RL Hyperparameter | Pure Shaping | Mixed Pool |
|---|---|---|
| algorithm | PPO | PPO |
| ppo_nminibatches | 2 | 2 |
| ppo_nepochs | 4 | 4 |
| ppo_clipping_epsilon | 0.2 | 0.2 |
| value_coefficient | 0.5 | 0.5 |
| clip_value | True | True |
| entropy_reg | 0 | 0 |
| advantage_normalization | False | False |
| reward_rescaling | 0.05 | 0.05 |
| $\gamma$ | 1 | 1 |
| $\lambda_{\text{td}}$ | 1 | 1 |
| $\lambda_{\text{gae}}$ | 1 | 1 |
| optimizer | ADAM | ADAM |
| adam_epsilon | 0.00001 | 0.00001 |
| learning_rate | 0.0003 | 0.0003 |
| max_grad_norm | 1 | 1 |

**Table 5:** Cleanup hyperparameters

| Hyperparameter fixed for the Cleanup experiments | Pure Shaping | Mixed Pool |
|---|---|---|
| training_iteration | 3000 | 30000 |
| meta_batch_size | 512 | 512 |
| batch_size (B) | 32 | 64 |
| num_inner_episode (M) | 100 | 5 |
| inner_episode_length (T) | 64 | 64 |
| p_naive | 1. | 0.75 |
| population_size (meta) | 1 | 3 |
| population_size (naive) | 10 | 3 |
| dynamic_naive_agents | False | True |

**Table 6:** The range of values swept over for hyperparameter search for each method for the Cleanup environment

| RL Hyperparameter | Pure Shaping | Mixed Pool |
|---|---|---|
| advantages_normalization | $\{False, True\}$ | $\{False, True\}$ |
| value_discount ($\gamma$) | $\{0.999, 1.0\}$ | $\{1.0\}$ |
| learning_rate | $\{0.003, 0.001, 0.0003\}$ | $\{0.03, 0.01, 0.5, 1.0\}$ |
| optimizer | $\{ADAM\}$ | $\{SGD\}$ |

**Table 7:** Hyperparameters used for the Cleanup Shaping and Cleanup Pool experiments.

| RL Hyperparameter | Cleanup Shaping | | | | Cleanup Pool | | | |
|---|---|---|---|---|---|---|---|---|
| | Coala | Batch Unaware | M-FOS | LOLA | Coala | Batch Unaware | M-FOS | LOLA |
| algorithm | PPO | PPO | PPO | - | A2C | A2C | A2C | - |
| ppo_nminibatches | 2 | 2 | 2 | - | - | - | - | - |
| ppo_nepochs | 4 | 4 | 4 | - | - | - | - | - |
| ppo_clipping_epsilon | 0.2 | 0.2 | 0.2 | - | - | - | - | - |
| value_coefficient | 0.5 | 0.5 | 0.5 | - | - | - | - | - |
| clip_value | True | True | True | - | True | True | True | True |
| entropy_regularization | 0 | 0 | 0 | 0 | 0 | 0 | 0 | 0 |
| advantage_normalization | True | True | True | True | True | True | True | True |
| $\gamma$ | 1 | 1 | 1 | 1 | 1 | 1 | 1 | 1 |
| $\lambda_{\text{td}}$ | 1 | 1 | 1 | 1 | 1 | 1 | 1 | 1 |
| reward_rescaling | 0.1 | 1 | 1 | 1 | 0.1 | 0.1 | 0.1 | 0.1 |
| $\lambda_{\text{gae}}$ | 1 | 1 | 1 | 1 | 1 | 1 | 1 | 1 |
| optimizer | ADAM | ADAM | ADAM | SGD | SGD | SGD | SGD | SGD |
| adam_epsilon | 0.00001 | 0.00001 | 0.00001 | - | - | - | - | - |
| learning_rate | 0.001 | 0.001 | 0.003 | 0.1 | 0.1 | 0.03 | 0.03 | 0.03 |
| max_grad_norm | 1 | 1 | 1 | - | 1 | 1 | 1 | 1 |

**Table 8:** Naive agent hyperparameters used across different settings

| RL Hyperparameter | IPD Shaping | IPD Mixed | Cleanup Shaping | Cleanup Mixed |
|---|---|---|---|---|
| algorithm | A2C | A2C | A2C | A2C |
| advantages_normalization | True | True | True | True |
| reward_rescaling | 0.05 | 0.05 | 0.1 | 0.1 |
| value_discount ($\gamma$) | 0.99 | 0.99 | 0.99 | 1 |
| td_lambda ($\lambda_{\text{td}}$) | 1.0 | 1.0 | 1.0 | 1.0 |
| gae_lambda ($\lambda_{\text{gae}}$) | 1.0 | 1.0 | 1.0 | 1.0 |
| value_coefficient | 0.5 | 0.5 | 0.5 | 0.5 |
| entropy_reg | 0.0 | 0.0 | 0.0 | 0.0 |
| optimizer | ADAM | ADAM | ADAM | SGD |
| adam_epsilon | 0.00001 | 0.00001 | 0.00001 | — |
| learning_rate | 0.005 | 0.005 | 0.005 | 1. |
| max_grad_norm | 1.0 | 1.0 | 1.0 | 1.0 |

## C  THE ANALYTICAL ITERATED PRISONER'S DILEMMA

For the experiments in Section 4 and 4.3, we analytically compute the discounted expected return of an infinitely iterated prisoner's dilemma, and its parameter gradients. Automatic differentiation allows us then to explicitly backpropagate through the learning trajectory of naive learners, to compute the ground-truth meta update. In the following, we provide details on this approach.

For both the naive learners and learning-aware meta agents, we consider tabular policies $\phi^i$ taking into account the previous action of both agents:

$$\phi^i = [\phi^i_0, \phi^i_1, \phi^i_2, \phi^i_3, \phi^i_4]^\top$$

with $\sigma(\phi^i_0)$ the probability of cooperating in the initial state (with sigmoid $\sigma$), and the next 4 parameters the logits of cooperating in states CC, CD, DC and DD respectively (CD indicates that first agent cooperated, and the second agent defected). As we use a tabular policy for the meta agents, they cannot accurately infer the opponent's parameters from context, but its policy gradient updates still inform it regarding the learning behavior of naive learners. Hence the meta agent can learn to shape naive learners while using a tabular policy, for example through zero-determinant extortion strategies (Press & Dyson, 2012). Using both policies, we can construct a Markov matrix providing the transition probabilities of one state to the next, ignoring the initial state.

$$M =$$

$$\left[ \sigma(\phi^1_{1:4}) \odot \sigma(\phi^2_{1:4}), \sigma(\phi^1_{1:4}) \odot (1 - \sigma(\phi^2_{1:4})), (1 - \sigma(\phi^1_{1:4})) \odot \sigma(\phi^2_{1:4}), (1 - \sigma(\phi^1_{1:4})) \odot (1 - \sigma(\phi^2_{1:4})) \right]^\top$$

with $\odot$ the element-wise product. Given the payoff vectors $r^1 = [1, -1, 2, 0]$ and $r^2 = [1, 2, -1, 0]$, and initial state distribution $s_0 = [\sigma(\phi^1_0)\sigma(\phi^2_0), \sigma(\phi^1_0)(1-\sigma(\phi^2_0)), (1-\sigma(\phi^1_0))\sigma(\phi^2_0), (1-\sigma(\phi^1_0))(1-\sigma(\phi^2_0))]^\top$ we can write the expected discounted return of agent $i$ as

$$J^i(\phi^1, \phi^2) = r^{i,\top} \left[ \sum_{t=0}^{\infty} \gamma^t M^t s_0 \right] \tag{15}$$

This discounted infinite matrix sum is a Neumann series of the inverse $(I - \gamma M)^{-1}$ with $I$ the identity matrix. This gives us:

$$J^i(\phi^1, \phi^2) = r^{i,\top} (I - \gamma M)^{-1} s_0 \tag{16}$$

Both $M$ and $s_0$ depend on the agent's policies, and we can compute the analytical gradients using automatic differentiation (we use JAX).

We model naive learners $\phi^{-i}$ as taking gradient steps on $J^{-i}$ with learning rate $\eta_{\text{naive}}$. The co-player shaping objective for meta agent $i$ is now

$$\bar{J}(\phi^i) = \sum_{m=0}^{M} J^i \left( \phi^i, \phi^{-i} + \sum_{q=1}^{m} \Delta_q \phi^{-i} \right) \quad \text{s.t.} \quad \Delta_q \phi^{-i} = \eta_{\text{naive}} \frac{\partial}{\partial \phi^{-i}} J^i \left( \phi^i, \phi^{-i} + \sum_{q'=1}^{q-1} \Delta_{q'} \phi^{-i} \right) \tag{17}$$

When a learning-aware meta agent faces a naive learner, we compute the shaping gradient by explicitly backpropagating through $\bar{J}(\phi^i)$, using automatic differentiation. When a learning-aware meta agent faces another meta agent, we compute the policy gradient as the partial gradient on $J^i(\phi^i, \phi^{-i})$, as with tabular policies, the meta agents deploy the same policy in each inner episode, and hence averaging over inner episodes is equivalent to playing a single episode of meta vs meta. For training the meta agents, we use a convex mixture of the gradients against naive learners and gradients against the other meta agent, with mixing factor $p_{\text{naive}}$. For the gradients against naive learners, we use a batch of randomly initialized naive learners of size `metabatch`. We use the adamw optimizer from the Optax library to train the meta agents, with default hyperparameters and learning rate $\eta_{\text{meta}}$.

For the LOLA experiments of Section 4.3, we compute the ground-truth LOLA-DICE updates equation 4 by initializing a naive learner with the opponent's parameters, simulate $M$ naive updates (look-aheads) following partial derivatives of $J^i$, and backpropagating through the final return $J^i \left( \phi^i, \phi^{-i} + \sum_{q=1}^{M} \Delta_q \phi^{-i} \right)$, including backpropagating through the learning trajectory. For Fig. 4, we train two separate LOLA agents against each other and report the training curves of the first agent (the training curves of the second agent are similar, data not shown). Using self-play

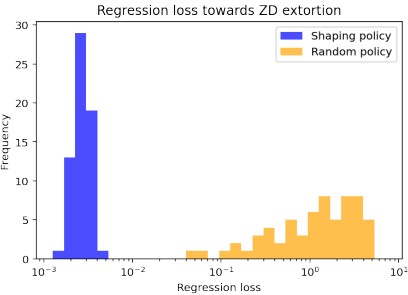

**Figure 8:** Histogram of the regression losses after fitting the $(\chi, \phi)$ parameters to the learned co-player shaping policies from Figure 3A for 64 random seeds, versus fitting the $(\chi, \phi)$ parameters to 64 uniform random policies.

instead of other-play resulted in similar results with the same main conclusions (data not shown). For all experiments, we used the following hyperparemeters: $\gamma = 0.95$, $\eta_{\text{meta}} = 0.005$, $\eta_{\text{naive}} = 5$ (except for 1-look-ahead, where we used $\eta_{\text{naive}} = 10$). For Figure 4C, we used a convex mixture of the LOLA-DICE gradient and the partial gradient on $J^i(\phi^i, \phi^{-i})$ with mixing factor $p_{\text{naive}}$. We used $p_{\text{naive}} = 1, 1, 0.75, 0.6, 0.4$ for look-aheads 1, 2, 3, 10 and 20 respectively.

## C.1 ADDITIONAL RESULTS ON THE ANALYTICAL IPD

**Learning-aware agents extort naive learners following Zero-Determinant-like extortion strategies.** Figure 3A shows that learning-aware agents trained against naive learners find a policy that extorts the naive learners into unfair cooperation. Here, we investigate the resulting extortion policies in more detail, and show that they are similar to the Zero-Determinant extortion strategies discovered by Press & Dyson (2012). Zero-determinant extortion strategies are parameterized by $\chi$ and $\phi$ as follows (with $(T, R, P, S) = (2, 1, 0, -1)$ the rewards of the prisoner's dilemma):

$$
\begin{aligned}
p_1 &= 1 - \phi(\chi - 1)\frac{R - P}{P - S} \\
p_2 &= 1 - \phi\left(1 + \chi\frac{T - P}{P - S}\right) \\
p_3 &= \phi\left(\chi + \frac{T - P}{P - S}\right) \\
p_4 &= 0
\end{aligned}
\tag{18}
$$

with $\chi \geq 1$ and $0 < \phi \leq \frac{P-S}{(P-S)+\chi(T-P)}$. For $\chi = 1$ and $\phi = \frac{P-S}{(P-S)+(T-P)}$ we recover the tit-for-tat strategy, representing the *fair* shaping strategy, whereas for higher values of $xi$, the resulting policies extort the naive learner into unfair cooperation. Note that Press & Dyson (2012) did not consider a $p_0$ parameter, as there theory is independent of the choice for $p_0$.

To investigate whether our learned co-player shaping policies are related to ZD extortion strategies, we take the converged policy $\sigma(\phi^i)$ after training with the pure shaping objective (c.f. Figure 3A), and fit the parameters $(\chi, \phi)$ to the regression loss $\|\sigma(\phi^i)[1:5] - p^{ZD}(\chi, \phi)\|^2$, with $p^{ZD}(\chi, \phi)$ the ZD extortion policy of Eq. 18. Figure 8 shows that policies learned with the co-player shaping objective can be well aproximated by ZD extortion policies, whereas random policies cannot. The ZD extortion policies of Eq. 18 consider undiscounted infinitely repeated matrix games, whereas we consider discounted infinitely repeated prisoner's dilemma with discount $\gamma = 0.999$. Furthermore, our shaping objective considers the cumuluted returns over the whole learning trajectory of the naive learner, in contrast to ZD extortion strategies that are optimized for the maximizing the return of the last inner episode. Hence, we should not expect an exact match between the learned policies $\sigma(\phi^i)$ and the ZD extortion strategies.

**Mutual unconditional defection is not a Nash equilibrium in the mixed group setting.** First, we check numerically whether mutual unconditional defection results in a zero gradient, a necessary condition for being a Nash equilibrium. As a zero probability corresponds to infinite logits, we

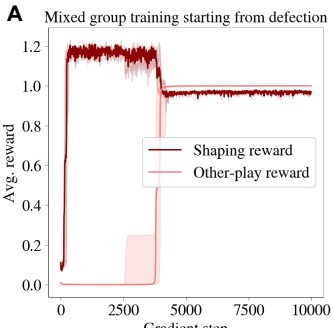 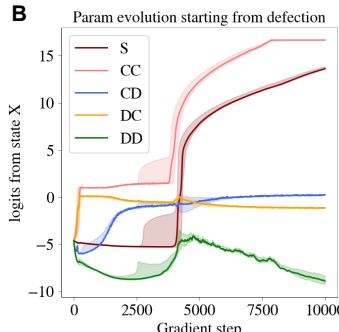

**Figure 9:** (A) Average reward during training in a mixed group setting with both agents starting from an unconditional defection policy, when evaluating the learned policy versus naive agents (shaping reward) and versus the other learned policy (other-play reward). (B) Parameter trajectory in logit space of the first agent (the second agent has similar learning trajectories, data not reported). Shaded regions indicate 0.25 and 0.75 quantiles, and solid lines the median over 8 random seeds.

parameterize our policy now directly in the probability space instead of logit space, and consider projected gradient ascent to the probability simplex, i.e. clipping the updated parameters between 0 and 1. For non-zero mixing factors $p_{\text{naive}}$, this results in a projected gradient that is 0 everywhere, except for the parameter corresponding to the DC state. For shaping naive learners, it is beneficial to reward co-players that cooperate by also cooperating with non-zero probability afterwards. Hence, when an agent with a pure defection policy plays against naive learners, the resulting gradient will push it out of the pure defection policy.

Figure 9A shows that when we train unconditional defection policies in the mixed group setting with the same hyperparameters as for Figure 3C, the agents escape mutual defection and learn to cooperate. Note that the agents quickly learn how to shape naive learners, and that it takes a bit longer to learn full cooperation, as the shaping objective does not provide pressure to increase the cooperation probability in the starting state. However, as the shaping policies of the meta agents are not any longer unconditional defection, playing against other meta agents provides a pressure to increase the cooperation probability, eventually leading to a phase transition towards cooperation. Figure 9B shows the parameter trajectory in logit space over training, showing that indeed the agents adjust quickly their parameters for shaping, and eventually also the initial state cooperation probability, leading to cooperation against other meta agents. As our policies are parameterized in logit space, we initialize them to $\log 0.01$ instead of exactly to zero cooperation probability to avoid infinities.

**Tit for Tat is not a Nash equilibrium in the mixed group setting.** Figure 10 repeats the same analysis but now starting from Tit for Tat policies, showing that mutual Tit for Tat is not a Nash equilibrium in our mixed pool setting. As we show in Figure 10C, this is caused by the possibility to shape naive learners faster by deviating from a strict tit for tat policy. Note that even though the resulting policies are not perfect tit for tat, they still fully cooperate when played against each other.

## D  PROOFS

**Theorem D.1.** *Take the expected shaping return* $\bar{J}(\phi^i) = \mathbb{E}_{\bar{P}^{\phi^i}}\left[\frac{1}{B}\sum_{b=1}^{B}\sum_{l=0}^{MT} R_l^{i,b}\right]$, *with* $\bar{P}^{\phi^i}$ *the distribution induced by the environment dynamics* $\bar{P}_t$, *initial state distribution* $\bar{P}_i$ *and policy* $\phi^i$. *Then the policy gradient of this expected return is equal to*

$$\nabla_{\phi^i}\bar{J}(\phi^i) = \mathbb{E}_{\bar{P}^{\phi^i}}\left[\sum_{b=1}^{B}\sum_{l=1}^{MT}\nabla_{\phi}\log\pi^i(a_l^{i,b}\mid h_l^{i,b})\left(\frac{1}{B}\sum_{l'=l}^{m_lT}r_{l'}^{i,b} + \frac{1}{B}\sum_{b'=1}^{B}\sum_{l'=m_lT+1}^{MT}r_{l'}^{i,b'}\right)\right]. \quad (19)$$

*Proof.* In the co-player shaping batched POMDP there is only one agent that is relevant for the policy gradient, as all other agents are naive learners and subsumed in the environment dynamics. Hence, to avoid overloading notations, we drop the $i$ superscript in the parameters, actions, policy and histories. Furthermore, we use the notation $a_l = \{a_l^b\}_{b=1}^{B}$ and similarly for $h_l$.

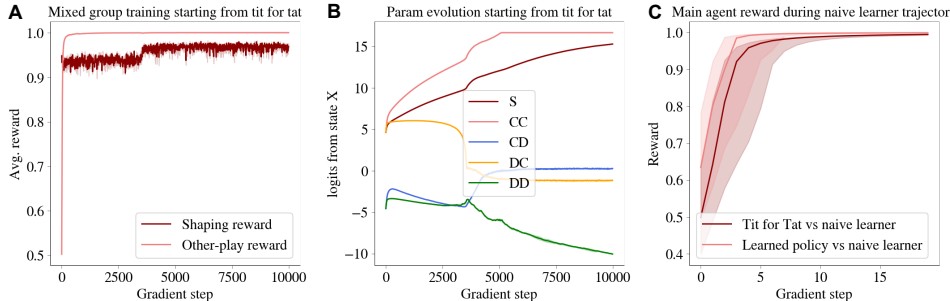

**Figure 10:** (A) Average reward during training in a mixed group setting with both agents starting from a tit for tat policy, when evaluating the learned policy versus naive agents (shaping reward) and versus the other learned policy (other-play reward). (B) Parameter trajectory in logit space of the first agent (the second agent has similar learning trajectories, data not reported). (C) The reward of a main agent when playing against a naive learner over a trajectory of 20 naive learning steps, showing that the policy learned after convergence in the mixed group setting shapes a naive learner faster compared to a tit for tat policy. Shaded regions indicate 0.25 and 0.75 quantiles, and solid lines the median over 8 random seeds.

We start by writing down the gradient of $\bar{J}(\phi) = \mathbb{E}_{\bar{P}^{\phi^i}}\left[\frac{1}{B}\sum_{b=1}^{B}\sum_{l=0}^{MT} R_l^{i,b}\right]$, making the summations in its expectation explicit.

$$\nabla_\phi \bar{J}(\phi) = \nabla_\phi \sum_{l=0}^{MT}\sum_{r_l\in\bar{\mathcal{R}}}\sum_{a_l\in\bar{\mathcal{A}}}\sum_{h_l\in\bar{\mathcal{H}}_l} \bar{P}^\phi(h_l)\bar{\pi}(a_l\mid h_l)\bar{P}(r_l\mid h_l,a_l)\frac{1}{B}\sum_b r_l^b$$

with $\bar{\mathcal{R}} = \times_b \mathcal{R}$ the joint reward space, $\bar{\mathcal{H}}_l$ the joint space over possible batched histories up until timestep $l$, and $\bar{\pi}(a_l\mid h_l) = \prod_{b=1}^B \pi(a_l^b\mid h_l^b)$. Applying the chain rule leads to

$$\nabla_\phi \bar{J}(\phi) = \sum_{l=0}^{MT}\sum_{r_l\in\bar{\mathcal{R}}}\sum_{a_l\in\bar{\mathcal{A}}}\sum_{h_l\in\bar{\mathcal{H}}_l} \nabla_\phi\bar{P}^\phi(h_l)\bar{\pi}(a_l\mid h_l)\bar{P}(r_l\mid h_l,a_l)\frac{1}{B}\sum_b r_l^b \dots$$
$$+ \sum_{l=0}^{MT}\sum_{r_l\in\bar{\mathcal{R}}}\sum_{a_l\in\bar{\mathcal{A}}}\sum_{h_l\in\bar{\mathcal{H}}_l} \bar{P}^\phi(h_l)\nabla_\phi\bar{\pi}(a_l\mid h_l)\bar{P}(r_l\mid h_l,a_l)\frac{1}{B}\sum_b r_l^b$$

as the reward dynamics $\bar{P}(r_l\mid h_l)$ are independent of the policy parameterization $\phi$. We first investigate the gradient of the marginal distribution $\nabla_\phi\bar{P}^\phi(h_l)$, by marginalizing over the joint trajectory distribution.

$$\nabla_\phi\bar{P}^\phi(h_l) = \nabla_\phi \sum_{\{a_{l'}\in\bar{\mathcal{A}},\{h_{l'}\in\bar{\mathcal{H}}_{l'}\}_{l'<l}\}} \bar{P}(h_l\mid h_{l-1},a_{l-1})\prod_{l'<l}\bar{P}(h_{l'}\mid h_{l'-1},a_{l'-1})\bar{\pi}(a_{l'}\mid h_{l'})$$
$$= \sum_{\{a_{l'}\in\bar{\mathcal{A}},\{h_{l'}\in\bar{\mathcal{H}}_{l'}\}_{l'<l}\}} \bar{P}(h_l\mid h_{l-1},a_{l-1})\prod_{l'<l}\bar{P}(h_{l'}\mid h_{l'-1},a_{l'-1})\bar{\pi}(a_{l'}\mid h_{l'})\sum_{l''<l}\nabla_\phi\log\bar{\pi}(a_{l''}\mid h_{l''})$$
$$= \mathbb{E}_{\bar{P}^\phi}\left[\sum_{l'<l}\nabla_\phi\log\bar{\pi}(a_{l'}\mid h_{l'})\right]$$

with $\{a_{l'}\in\bar{\mathcal{A}},\{h_{l'}\in\bar{\mathcal{H}}_{l'}\}_{l'<l}$ the joint space over all actions and histories over timesteps $l' < l$. In the second line, we used the chain rule and $\nabla_\phi\bar{\pi} = \bar{\pi}\nabla_\phi\log\bar{\pi}$. In the third line we renamed the index $l''$ to $l'$.

Filling this expression for $\nabla_\phi\bar{P}^\phi(h_l)$ into our expression for $\nabla_\phi\bar{J}(\phi)$, combined with the log trick $\nabla_\phi\bar{\pi} = \bar{\pi}\nabla_\phi\log\bar{\pi}$ and using the expectation notation for clarity of notation, we end up with

$$\nabla_\phi\bar{J}(\phi) = \mathbb{E}_{\bar{P}^\phi}\left[\frac{1}{B}\sum_{b=1}^{B}\sum_{l=0}^{MT}r_l^b\sum_{l'\leq l}\nabla_\phi\log\bar{\pi}(a_{l'}\mid h_{l'})\right]$$

Reordering the summations, and using $\bar{\pi}(a_l \mid h_l) = \prod_{b=1}^{B} \pi(a_l^b \mid h_l^b)$ leads to

$$\nabla_\phi \bar{J}(\phi) = \mathbb{E}_{\bar{P}^\phi}\left[\sum_{b=1}^{B}\sum_{l=1}^{MT} \nabla_\phi \log \pi(a_l^b \mid h_l^b)\left(\frac{1}{B}\sum_{b'=1}^{B}\sum_{l'=l}^{MT} r_{l'}^{b'}\right)\right].$$

Finally, actions can only influence the other parallel trajectories through the parameter updates of the naive learners in the environment, which takes place at the inner episode boundaries. Hence, during the current inner episode (before a naive learner update takes place), rewards $r_l^{b'}$ are independent of actions $a_l^b$ for $b \neq b'$. As $\mathbb{E}_{a\sim\pi}[c \log \pi(a \mid h)] = 0$ for a constant $c$ independent of the actions, the policy gradient is equal to

$$\nabla_\phi \bar{J}(\phi) = \mathbb{E}_{\bar{P}^\phi}\left[\sum_{b=1}^{B}\sum_{l=1}^{MT} \nabla_\phi \log \pi(a_l^b \mid h_l^b)\left(\frac{1}{B}\sum_{l'=l}^{m_l T} r_{l'}^b + \frac{1}{B}\sum_{b'=1}^{B}\sum_{l'=m_l T+1}^{MT} r_{l'}^{b'}\right)\right].$$

with $m_l$ the inner episode index corresponding to timestep $l$, thereby concluding the proof.

$\square$

**Theorem D.2.** *Assuming that (i) the COALA policy is only conditioned on inner episode histories $x_{m,t}^{i,b}$ instead of long histories $h_l^{i,b}$, and (ii) the naive learners are initialized with the current parameters $\phi^{-i}$ of the other agents, then the* COALA-PG *update on the batched co-player shaping POMDP is equal to*

$$\nabla_{\phi^i}\bar{J}(\phi^i) = \frac{\mathrm{d}}{\mathrm{d}\phi^i}\sum_{m=0}^{M}\mathbb{E}_{\bar{P}^{\phi^i}}\left[J^i\left(\phi^i, \phi^{-i} + \sum_{q=1}^{m}\Delta\phi^{-i}(\bar{x}_{q,T})\right)\right] \tag{20}$$

*with $\Delta\phi^{-i}(\bar{x}_{q,T})$ the naive learner's update based on the batch of inner-episode histories $\bar{x}_{q,T}$, and $\frac{\mathrm{d}}{\mathrm{d}\phi^i}$ the total derivative, taking into account both the influence of $\phi^i$ through the first argument of $J^i(\phi^i, \phi^{-i} + \sum_{q=1}^{m}\Delta\phi^{-i}(\bar{x}_{q,T}))$, as well as through the parameter updates $\Delta\phi^{-i}$ by adjusting the distribution over $\bar{x}_{q,T}$.*

*Proof.* We start by restructuring the long-history expected return $\bar{J}$ of the batched co-player shaping POMDP into a sum of inner episode expected returns $J$ from the multi-agent POSG, leveraging the assumptions that (i) the COALA policy is only conditioned on inner episode histories $x_{m,t}^{i,b}$ instead of long histories $h_l^{i,b}$, and (ii) the naive learners are initialized with the current parameters $\phi^{-i}$ of the other agents.

$$\begin{aligned}
\bar{J}(\phi^i) &= \mathbb{E}_{\bar{P}^{\phi^i}}\left[\frac{1}{B}\sum_{b=1}^{B}\sum_{t=0}^{MT} R_t^{i,b}\right]\\
&= \sum_{m=0}^{M-1}\mathbb{E}_{\bar{P}^{\phi^i}}\left[\frac{1}{B}\sum_{b=1}^{B}\sum_{t=mT+1}^{(m+1)T} R_t^{i,b}\right]\\
&= \sum_{m=0}^{M-1}\mathbb{E}_{\bar{P}^{\phi^i}(\bar{h}_{mT})}\left[\frac{1}{B}\sum_{b=1}^{B}\mathbb{E}_{P^{\phi^i,\phi^{-i}+\sum_{q=1}^{m}\Delta\phi^{-i}(\bar{x}_{q,T})}}\left[\sum_{t=mT+1}^{(m+1)T} R_t^{i,b}\right]\right]\\
&= \sum_{m=0}^{M-1}\mathbb{E}_{\bar{P}^{\phi^i}}\left[J^i\left(\phi^i, \phi^{-i} + \sum_{q=1}^{m}\Delta\phi^{-i}(\bar{x}_{q,T})\right)\right]
\end{aligned}$$

with $P^{\phi^i,\phi^{-i}+\sum_{q=1}^{m}\Delta\phi^{-i}(\bar{x}_{q,T})}$ the distribution induced by the environment dynamics of the multi-agent POSG, played with policies $(\phi^i, \phi^{-i} + \sum_{q=1}^{m}\Delta\phi^{-i}(\bar{x}_{q,T}))$. The step from line two to three is made possible by the assumption that the COALA policy $\pi^i$ is only conditioned on inner episode histories $x_{m,t}^{i,b}$ instead of long histories $h_l^{i,b}$. This ensures that the distribution $\bar{P}^{\phi^i}$ over the batch of inner-episode histories $\bar{x}_{m+1,T}$ only depends on $(\phi^i, \phi^{-i} + \sum_{q=1}^{m}\Delta\phi^{-i}(\bar{x}_{q,T}))$, as the policy $\pi^i$ does not take previous observations from before the current inner episode boundary into account.

A policy gradient takes into account the full effect of the parameters of a policy on the trajectory distribution induced by the policy. In our batched co-player shaping POMDP, the trajectory distribution induced by policy $\phi^i$ influences the reward distribution in the concatenated POSG, as well as the naive learner's current parameters through the inner episode batches they use for their updates. Hence, we have that

$$\nabla_{\phi^i} \bar{J}(\phi^i) = \frac{\mathrm{d}}{\mathrm{d}\phi^i} \sum_{m=0}^{M} \mathbb{E}_{\bar{P}^{\phi^i}} \left[ J^i \left( \phi^i, \phi^{-i} + \sum_{q=1}^{m} \Delta\phi^{-i}(\bar{x}_{q,T}) \right) \right]$$

with $\frac{\mathrm{d}}{\mathrm{d}\phi^i}$ the total derivative, taking into account both the influence of $\phi^i$ through the first argument of $J^i(\phi^i, \phi^{-i} + \sum_{q=1}^{m} \Delta\phi^{-i}(\bar{x}_{q,T}))$, as well as through the parameter updates $\Delta\phi^{-i}$ by adjusting the distribution over $\bar{x}_{q,T}$, thereby concluding the proof.

$\square$

# E    RELATING COALA-PG TO LEARNING WITH OPPONENT-LEARNING AWARENESS (LOLA)

In this section, we establish a formal relationship between COALA-PG and Learning with Opponent-Learning Awareness (LOLA; Foerster et al., 2018a), the seminal work that spearheaded the learning awareness field. In doing so, we further show how COALA-PG can be used to derive a new LOLA gradient estimator that does not require higher-order derivative estimates.

LOLA considers a POSG (c.f. Section 2.2) with agent policies $(\phi^i, \phi^{-i})$, and expected return $J^i(\phi^i, \phi^{-i})$. Recall from Section 4 that instead of estimating the naive gradients $\nabla_{\phi^i} J^i(\phi^i, \phi^{-i})$, LOLA anticipates that co-players update their parameters with $M$ naive gradient steps. The improved LOLA-DICE (Foerster et al., 2018b) update reads:

$$\nabla_{\phi^i}^{\mathrm{LOLA}} = \frac{\mathrm{d}}{\mathrm{d}\phi^i} \left[ J^i \left( \phi^i, \phi^{-i} + \sum_{q=1}^{M} \Delta_q \phi^{-i} \right) \quad \text{s.t.} \ \Delta_q \phi^{-i} = \alpha \frac{\partial}{\partial\phi^{-i}} J^i \left( \phi^i, \phi^{-i} + \sum_{q'=1}^{q-1} \Delta_{q'} \phi^{-i} \right) \right] \tag{21}$$

with $\frac{\mathrm{d}}{\mathrm{d}\phi^i}$ the total derivative taking into account the effect of $\phi^i$ on the parameter updates $\Delta_q \phi^{-i}$, and $\frac{\partial}{\partial\phi^{-i}}$ the partial derivative.

Despite the apparent dissimilarities of the two algorithms, we now show in Theorem E.1 that as a special case of `COALA-PG`, we can estimate the gradient of a similar objective to the LOLA-DICE method. We consider a mixed agent group of meta agents $(\phi^i, \phi^{-i})$, and naive learners.

**Theorem E.1.** *Assuming that (i) the COALA policy is only conditioned on inner episode histories $x_{m,t}^{i,b}$ (instead of long histories $h_l^{i,b}$), with subscript $m = 1, \ldots, M$ indexing histories over meta-steps, and (ii) the naive learners are initialized with the current parameters $\phi^{-i}$ of the other agents, then the `COALA-PG` update on the batched co-player shaping POMDP equals*

$$\nabla_{\phi^i} \bar{J}(\phi^i) = \frac{\mathrm{d}}{\mathrm{d}\phi^i} \sum_{m=0}^{M} \mathbb{E}_{\bar{P}^{\phi^i}} \left[ J^i \left( \phi^i, \phi^{-i} + \sum_{q=1}^{m} \Delta\phi^{-i}(\bar{x}_{q,T}) \right) \right] \tag{22}$$

*with $\Delta\phi^{-i}(\bar{x}_{q,T})$ the naive learner's update based on the batch of inner-episode histories $\bar{x}_{q,T}$.*

Compared to the LOLA-DICE gradient of Eq. 21, there are two main differences. (i) LOLA-DICE assumes that the naive learner takes a deterministic gradient step on $J$, whereas `COALA-PG` takes into account that the naive learner takes a stochastic policy gradient step based on the current mini-batch of inner policy histories. When $J$ is linear, we can bring the expectation in the `COALA-PG` expression inside, resulting in the deterministic gradient, but in general $J$ is nonlinear. (ii) LOLA-DICE considers only the average inner episode return $J$ after $M$ naive learner updates, whereas `COALA-PG` considers the whole learning trajectory.

The above two differences are rooted in the distinction of the objectives on which LOLA-DICE and `COALA-PG` estimate the policy gradient. A bigger difference arises on *how* both methods estimate

the policy gradient. As LOLA-DICE assumes that the naive learner takes deterministic gradient updates, their resulting gradient estimator backpropagates explicitly through this learning update, resulting in higher-order derivative estimates. By contrast, `COALA-PG` assumes that the naive learner takes stochastic gradient updates, and can hence estimate the policy gradient through measuring the effect of $\phi^i$ on the distribution of $\bar{x}_{l,T}$, and thereby on the resulting co-player parameter updates, without requiring higher-order derivatives.

We emphasize that both estimators have their strengths and weaknesses. When LOLA-DICE has access to an accurate model of the co-players and their learning algorithm, explicitly backpropagating through this learning algorithm and model can provide detailed gradient information. However, when the co-player and learning algorithm model are inaccurate, the detailed higher-order derivative information can actually hurt (Zhao et al., 2022). In this case, a higher-order-derivative-free approach such as `COALA-PG` can be beneficial.

In sum, in contrast to existing methods, we introduce a return estimator that avoids higher-order derivative computations while taking into account that other agents are themselves undergoing mini-batched reinforcement learning. Furthermore, while methods of the LOLA type require an explicit model of the opponent and its update function, `COALA-PG` allows for more flexible modeling. Importantly, this enables exploiting the information in long histories that cover multiple inner episodes with a powerful sequence model, for which credit assignment can be carried out over long time scales. This makes it possible to combine implicit co-player modeling with modeling the *learning* of co-players, a process known as algorithm distillation (Laskin et al., 2022).

## F    Deriving prior shaping algorithms from Eq. 1

The general shaping problem due to Lu et al. (2022) presented in Eq. 1, which we repeat below for convenience, captures many of the relevant co-player shaping techniques in the literature.

$$\max_{\mu} \mathbb{E}_{\tilde{P}_i(\phi_0^i, \phi_0^{-i})} \mathbb{E}_{\tilde{P}^{\mu}} \left[ \sum_{m=m_{\text{start}}}^{M} J^i(\phi_m^i, \phi_m^{-i}) \right], \tag{23}$$

M-FOS solves this co-player shaping problem without making further assumptions (Lu et al., 2022). Good Shepherd uses a stateless meta-policy $\mu(\phi_m^i; \theta) = \delta(\phi_m^i = \theta)$, with $\delta$ the Dirac-delta distribution (Balaguer et al., 2022). Meta-MAPG (Kim et al., 2021) meta-learns an initialization $\phi_0^i = \theta$ in the spirit of model-agnostic meta-learning (Finn et al., 2017), and lets every player learn by following gradients on their respective objectives. The meta-value learning method (Cooijmans et al., 2023) models the meta-policy $\mu(\phi_{m+1}^i \mid \phi_m^i, \phi_m^{-i}; \theta)$ as the gradient on a meta-value function $V(\phi_m^i, \phi_m^{-i})$ parameterized by $\theta$. Finally, we recover a single LOLA update step (Foerster et al., 2018a) by initializing $(\phi_0^i, \phi_0^{-i})$ with the current parameters of all agents, taking $M = m_{\text{start}} = 1$, using a stateless meta-policy $\mu(\phi_m^i; \theta) = \delta(\phi_m^i = \theta)$ and taking a single gradient step w.r.t. $\theta$, instead of solving the shaping problem to convergence.

## G    Detail on baseline methods

### G.1    Batch-unaware COALA PG

We remind that the policy gradient expression for COALA is as follows

$$\nabla_{\phi^i} \bar{J}(\phi^i) = \mathbb{E}_{\bar{P}^{\phi^i}} \left[ \sum_{b=1}^{B} \sum_{l=1}^{MT} \nabla_{\phi^i} \log \pi^i(a^{i,b}l \mid h_l^{i,b}) \left( \frac{1}{B} \sum_{b'=1}^{B} \sum_{k=l}^{MT} R_k^{i,b'} \right) \right]. \tag{24}$$

The batch-unaware COALA PG is the naive baseline, consisting in applying policy gradient methods to individual trajectories in a batch, i.e.

$$\hat{\nabla}_{\phi^i \text{batch-unaware}} \bar{J}(\phi^i) = \mathbb{E}_{\bar{P}^{\phi^i}} \left[ \frac{1}{B} \sum_{b=1}^{B} \sum_{l=1}^{MT} \nabla_{\phi^i} \log \pi^i(a^{i,b}l \mid h_l^{i,b}) \sum_{k=l}^{MT} R_k^{i,b} \right]. \tag{25}$$

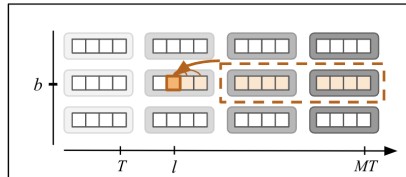

**Figure 11:** Diagram visualizing the difference between the COALA-PG update and the batch unaware COALA policy gradient update.

Figure 11 visualizes the main difference between the COALA-PG update and the batch-unaware COALA-PG update.

### G.2 M-FOS

Lu et al. (2022) consider both a policy gradient method and evolutionary search method to optimize the shaping problem of Eq. 1. Here, we focus on the M-FOS policy gradient method, which was used by Lu et al. (2022) in their Coin Game experiments, which is their only experiment that goes beyond tabular policies. M-FOS uses a distinct architecture with a recurrent neural network as inner policy, which receives an extra conditioning vector as input from a meta policy that processes inner episode batches. As shown by Khan et al. (2024), we can obtain better performance by combining both inner and meta policy in a single sequence model with access to the full history $h_l$ containing all past inner episodes. Hence, we use this improved architecture for our M-FOS baseline, which we train by the policy gradient method proposed by Lu et al. (2022). This allows us to use the same architecture for both M-FOS and COALA.

As the manuscript of (Lu et al., 2022) does not explicitly mention how to deal with the inner-batch dimension of a naive learner, we reconstruct the learning rule from their publicly available codebase. By denoting by $m_l$ the inner episode index corresponding to the meta episode time step $l$, the update is as follows:

$$\hat{\nabla}_{\phi^i \text{M-FOS}} \bar{J}(\phi^i) = \mathbb{E}_{\bar{P}^{\phi^i}} \left[ \sum_{b=1}^{B} \sum_{l=1}^{MT} \nabla_{\phi^i} \log \pi^i(a_l^{i,b} \mid h_l^{i,b}) \left( \sum_{k=l}^{Tm_l} R_k^{i,b} + \frac{1}{B} \sum_{b'=1}^{B} \sum_{k=Tm_l+1}^{MT} R_k^{i,b'} \right) \right]. \tag{26}$$

We note that when taking the inner-episode boundaries into account in the COALA-PG update, due to some terms disappearing from the expectation, the expression becomes

$$\nabla_{\phi^i} \bar{J}(\phi^i) = \mathbb{E}_{\bar{P}^{\phi^i}} \left[ \sum_{b=1}^{B} \sum_{l=1}^{MT} \nabla_{\phi^i} \log \pi^i(a_l^{i,b} \mid h_l^{i,b}) \left( \frac{1}{B} \sum_{k=l}^{Tm_l} R_k^{i,b} + \frac{1}{B} \sum_{b'=1}^{B} \sum_{k=Tm_l+1}^{MT} R_k^{i,b'} \right) \right]. \tag{27}$$

One can see that the M-FOS update rule closely resembles COALA-PG, except for the fact that it is lacking a factor $\frac{1}{B}$ in front of the reward of the current inner episode. As we show in our experiments, this biases the policy gradient and introduces inefficiencies in the optimization.

## H  ADDITIONAL EXPERIMENTAL RESULTS

### H.1 BALANCING GRADIENT CONTRIBUTIONS

The main difference between COALA-PG (c.f. Eq. 3) and M-FOS is the scaling of the current reward-episode. The main difference between COALA-PG equation 3 and M-FOS equation 7 is that COALA-PG scales the return of the current inner episode by $\frac{1}{B}$, whereas M-FOS does not. This scaling is crucial, as the the influence of an action on future inner episodes is of order $O(\frac{1}{B})$ because

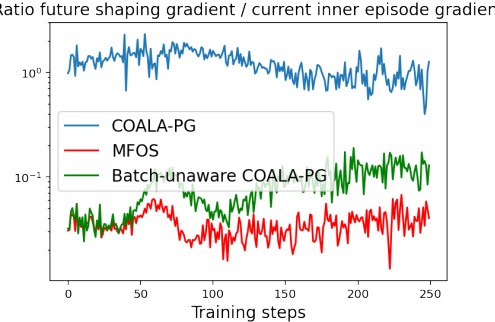

**Figure 12:** Y-axis: Ratio of the magnitude of the policy gradient contribution arising from future inner episode returns (the co-player shaping learning signal) w.r.t. the gradient contribution arising from the current inner episode return. X-axis: meta gradient steps on the iterated prisoner's dilemma, in a mixed group setting corresponding to the setting of Fig. 5C, and an increased meta-batch size of 2048 to reduce the variance of the gradient estimates.

the naive learner averages its update over the $B$ inner episode trajectories. Hence, by scaling the inner episode return by $\frac{1}{B}$, COALA-PG ensures that both contributions have the same scaling w.r.t. $B$, and finally by summing the policy gradient over the minibatch instead of averaging, we end up with a policy gradient of $O(1)$. M-FOS and batch-unaware COALA do not scale the current inner episode return, which causes the co-player shaping learning signal to vanish w.r.t. the current inner episode return, resulting in poor shaping performance.

Figure 12 confirms that empirically, COALA-PG correctly balances the gradient contributions from the current inner episode return with those from future inner episode returns. In contrast, M-FOS and batch-unaware COALA have unbalanced gradient contributions, with the co-player shaping gradient contribution vanishing w.r.t. the current inner episode return contribution.

## H.2 DETAILED RESULTS ON CLEANUP−LITE

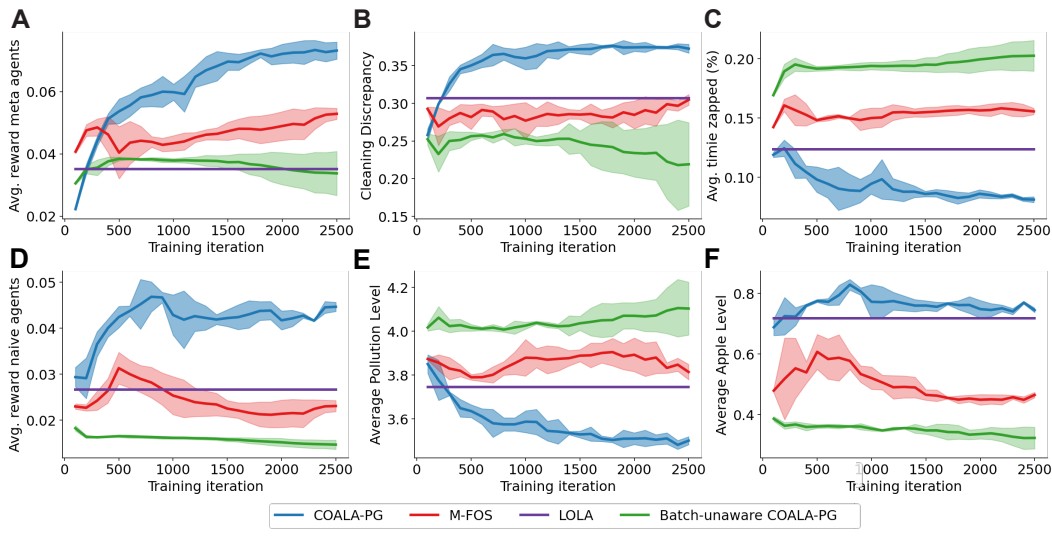

**Figure 13: Agents trained by `COALA-PG` against naive agents only successfully shape them in `CleanUp-lite`.** (A) `COALA-PG`-trained agents better shape naive opponents compared to baselines, obtaining higher return. (B and C) Analyzing behavior within a single meta-episode after training reveals that COALA outperforms baselines and shapes naive agents, (i) exhibiting a lower cleaning discrepancy (absolute difference in average cleaning time between the two agents), and (ii) being less often zapped. (D) Average reward of naive learners is higher when playing with `COALA-PG` agents compared to other agents. (E and F): `COALA-PG` results in lower average pollution level and higher average apple level. Shaded regions indicate standard deviation computed over 5 seeds.

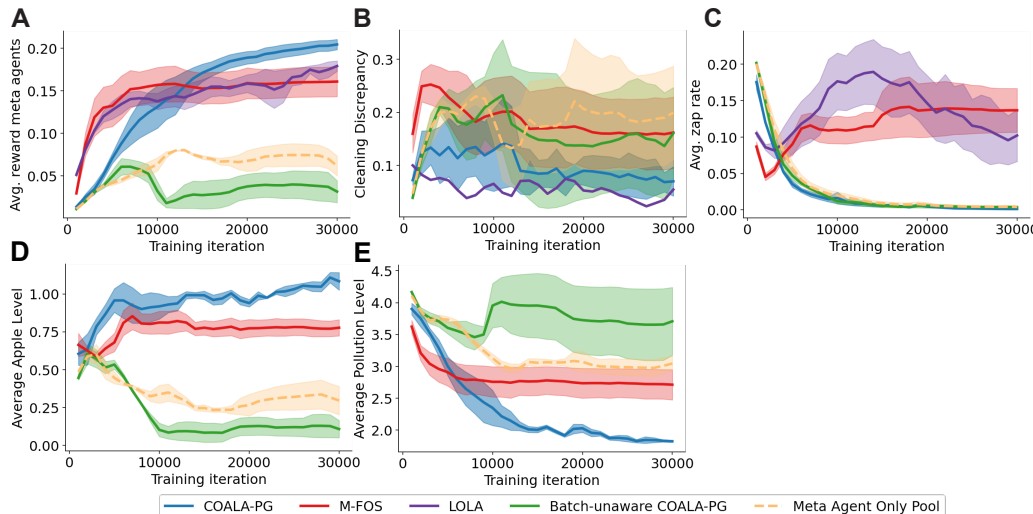

**Figure 14: Agents trained with `COALA-PG` against a mixture of naive and other meta agents learn to cooperate in `CleanUp-lite`**. (A) `COALA-PG`-trained agents obtain higher average reward than baseline agents when playing against each other. (B and C): `COALA-PG` leads to a more fair division of cleaning efforts and lower zapping rates. (D and E): `COALA-PG` results in lower average pollution level and higher average apple level. Shaded regions indicate standard deviation.

### H.3   TRAINING A `COALA-PG` AGENT VERSUS AN M-FOS AGENT ON ITERATED PRISONER'S DILEMMA

We investigate training a `COALA-PG` versus an M-FOS agent, our strongest performing meta-agent baseline. In this setup, we still use a mixture of naive learners and meta agents. Only now when a `COALA-PG` agent either plays against an M-FOS agent or naive learner, and an M-FOS agent either plays against a `COALA-PG` agent or a naive learner. We found that `COALA-PG` successfully shapes M-FOS into cooperation, reaching average rewards of $0.850 \pm 0.037$ for `COALA-PG` and $0.853 \pm 0.017$ for M-FOS. This is in stark contrast to when M-FOS agents only play against other M-FOS agents and naive learners, which converges to mutual defection.

We did not investigate training a `COALA-PG` agent versus a LOLA agent, as the training setup of both agents is fundamentally different, and would pose an unfair disadvantage for LOLA agents. A LOLA agent learns on the same timescale as naive learners, taking a few look-ahead steps into account in its update. Hence, the full learning trajectory of a LOLA agent is considered as one meta trajectory for a `COALA-PG` agent. As `COALA-PG` agents would play many different meta trajectories, always against a freshly initialized LOLA agent, this would give a `COALA-PG` detailed information about the learning behavior of LOLA agents, whereas LOLA agents are left in the dark as they cannot observe the *meta* updates of `COALA-PG`. Hence, `COALA-PG` would extort LOLA agents similar to naive learners.

## I   SOFTWARE

The results reported in this paper were produced with open-source software. We used the Python programming language together with the Google JAX (Bradbury et al., 2018) framework, and the NumPy (Harris et al., 2020), Matplotlib (Hunter, 2007), Flax (Heek et al., 2024) and Optax (Babuschkin et al., 2020) packages.

