# OpenReview forum: "Multi-agent cooperation through learning-aware policy gradients"
_ICLR.cc/2025/Conference — ICLR 2025 Poster_

### Official Review · Reviewer_UL6h · 2024-10-27

**Soundness:** 3
**Presentation:** 3
**Contribution:** 2
**Rating:** 6
**Confidence:** 3

**Summary:**

This work proposes a new paradigm for MARL that enables agents to be aware of inter-episode dynamics and presents compelling experimental results in both the iterated prisoner’s dilemma and a grid-world sequential social dilemma.

**Strengths:**

- This work addresses a central challenge in MARL.
- The proposed framework (Figures 1 and 2) is important.
- The method parameters are thoroughly detailed in the appendix.
- Finding 1 is especially intriguing.

**Weaknesses:**

- Line 114: The metric used in this work is ambiguous. Could you clarify which metric is applied in subsequent sections? What expectations should we have to claim that the proposed method outperforms other algorithms?
	- In Line 485, it appears the objective is solely the meta-agent's reward rather than equilibrium. Can we then say that this work is aimed at developing an algorithm to exploit naive agents?
	- Or is this an agent-based simulation method?

- Line 191: The definition states that the "state" in this game includes all agents' policy parameters. This creates an exponentially large space, especially when agents are assumed to use neural networks. In this case, the state encompasses the combined network parameters of all agents.
	- To implement a meta-agent in this setting, does the input of its policy network include the policy network parameters of other naive agents? If so, how is this optimized and implemented?
	- According to equation (4) and (5), I suppose that the meta agent does not input the others' parameters to its own policy network, but treats it by sampling the gradient. Then why is this estimation unbiased? Using sampling method does not seem to solve the non-stationarity issue. And this contradicts the motivation of this method.
	- If there are two meta-agents in the training phase, how do they treat each other? Are their gradient chains intertwined, or do they simply treat one another as naive agents?
- Line 998: The proof relies on Sutton’s policy gradient theorem, which assumes a fully observable environment. But Line 72 states that this environment is partially observable. Does the derivation still hold in this case? (I doubt the gradient computation is exact in this context, but I understand it may be an acceptable estimate for experimental purposes.)
	- Equation (4) shouldn’t be an equality but should have a scaling factor.
- Line 370: This experiment begins from a specific initialization point. The maximum expected reward is slightly lower than the cooperation reward of 1. Although this is a challenging case, what would results look like for different initialization points? Can a learning-aware agent achieve better outcomes (with an expected reward greater than 1)?
- Figure 3B and 3C: Does the proposed method require interaction with naive agents to initiate training?
- Why is there no comparison between COALA and M-FOS in Figure 5?

Minors:

- Why are all references in blue font? This formatting seems differ from other submissions.
- In Lines 14, 33, 49, 106, 122, 161, 185, 478, and 479, the single quotation marks should be replaced with double quotation marks.

**Questions:**

Questions are mentioned in the weaknesses part.

---

> ### Author Response · Authors · 2024-11-21
> **Rebuttal (Part 1)**
>
> We thank the reviewer for their thoughtful review, which helped us improve our work. We reply to the concerns and questions raised point-by-point below.
>
> > *Line 114: The metric used in this work is ambiguous. Could you clarify which metric is applied in subsequent sections? What expectations should we have to claim that the proposed method outperforms other algorithms?
> In Line 485, it appears the objective is solely the meta-agent's reward rather than equilibrium. Can we then say that this work is aimed at developing an algorithm to exploit naive agents?
> Or is this an agent-based simulation method?*
>
> We note that for the social dilemmas we focused on, equilibria with high sole agent rewards coincide with the highest global social welfare equilibria. As such, it is easy to define the metric, either using individual reward, or total rewards received by all involved meta agents. We nonetheless agree that this was unclear, and that it is in general not obvious how to define success in general-sum games.
>
> To clarify the points raised, we have implemented the following changes in the paper:
> * We clarify the metrics on the captions of Figs. 5 and 7. In general, for situations where a meta agent only plays against naive learners with the objective to shape them, we report the individual average reward of the meta agent. For situations where meta agents play against a mixed group of other meta agents and naive learners, we report the average global social welfare while evaluating meta agents against other meta agents in the group.
> * We now clearly state in the beginning of Section 5 that our aims are twofold: to understand (i) when meta-agents succeed in exploiting naive agents, and (ii) when cooperation is achieved among meta-agents. Our method outperforms other baselines on these two fronts: (i) it is more successful in exploiting naive agents, and (ii), when playing with other COALA agents, it achieves cooperative equilibria which Pareto dominates (achieves strictly more reward for all meta agents involved) equilibria found by the M-FOS and batch-unaware baselines.
>
>
> > *Line 191: The definition states that the "state" in this game includes all agents' policy parameters. This creates an exponentially large space, especially when agents are assumed to use neural networks. In this case, the state encompasses the combined network parameters of all agents. To implement a meta-agent in this setting, does the input of its policy network include the policy network parameters of other naive agents? If so, how is this optimized and implemented?*
>
> Our COALA method is grounded in a POMDP setting. Specifically, even though the hidden state of the environment includes all agents’ policy parameters, the policy of the meta agent is only conditioned on the observation history, which we design to not include the other agents’ policy parameters. This allows us to overcome the problem of requiring access to the other agents’ parameters, while still having access to principled policy gradients. Instead of explicitly providing the parameters of co-players to the meta-agent policy, the sequence model meta-learns to infer “in-context” the parameters and learning algorithms employed by co-players based solely on observations including their past actions. Our approach is similar in spirit to the classic single-agent meta reinforcement learning methods due to Wang et al., 2016, and Duan et al., 2017. This is in fact one of the key strengths of our method, in contrast to e.g. LOLA [Foerster et al. 2017], or some of the implementations in M-FOS, which require explicit access to co-player parameters.
>
> We will clarify this contrast in the final version of our manuscript.
>
> > *According to equation (4) and (5), I suppose that the meta agent does not input the others' parameters to its own policy network, but treats it by sampling the gradient. Then why is this estimation unbiased? Using sampling method does not seem to solve the non-stationarity issue. And this contradicts the motivation of this method.*
>
> As we constructed a correct POMDP formulation of the co-player shaping setup, we can follow standard RL practices to derive an unbiased policy gradient estimator corresponding to this POMDP, using Monte Carlo samples and value function approximators (c.f. Appendix A and D). Note that we neither require access to the co-players’ parameters, nor require to sample gradient updates of co-players. For full transparency, we have rewritten Theorem 3.1 and its proof from scratch, and in full detail, without invoking prior work (c.f. Appendix D). The new proof should make it clear that the estimator is indeed unbiased, and exact.

---

> > ### Author Response · Authors · 2024-11-21
> > **Rebuttal (Part 2)**
> >
> > > *If there are two meta-agents in the training phase, how do they treat each other? Are their gradient chains intertwined, or do they simply treat one another as naive agents?*
> >
> > The batched co-player-shaping POMDP defined in section 2.3, and the COALA policy gradient derivation of section 3 are defined for the setting where 1 meta agent plays against naive learner co-players. In this setting, the meta agents face a stationary single-agent POMDP, as the naive learners can be subsumed in a stationary environment including their learning dynamics.  In Section 4, we show how we can use the COALA-PG method for training multiple meta agents against each other and reach equilibria with higher global social welfare. When two meta agents play against each other during the training phase, it is again a standard multi-agent reinforcement learning setting. The main difference with the standard multi-agent RL setup is that now the meta agents play longer games, consisting of M concatenated inner episodes.  Both meta agents have a sequence model as policy and do not update their parameters over the course of a meta trajectory, only after the whole meta trajectory is completed. Hence, neither are the gradient chains of the meta agents intertwined, nor do they inconsistently treat one another as a naive agent that updates its parameters after each inner episode. However, in this setup, two meta agents playing against each other again represents a non-stationary setting. Importantly, as we show in detail in Section 4, and confirm empirically in Section 5, when meta agents play both against other meta agents, as well as naive learners, this mix of experience guides the learning of all involved meta agents towards equilibria with higher social welfare. Hence, even though the setting is again non-stationary, the mix of incentives of both shaping naive learners and playing well against other meta learners leads to better equilibria.
> >
> > > *Line 998: The proof relies on Sutton’s policy gradient theorem, which assumes a fully observable environment. But Line 72 states that this environment is partially observable. Does the derivation still hold in this case? (I doubt the gradient computation is exact in this context, but I understand it may be an acceptable estimate for experimental purposes.)
> > Equation (4) shouldn’t be an equality but should have a scaling factor.*
> >
> > The reviewer is correct in noting that Sutton’s policy gradient theorem applies to MDPs. However, our derivation still holds as the policy gradient theorem can readily be applied to POMDPs as well, by observing that any POMDP can be converted to a belief-state MDP, where the observation history is a sufficient statistic for the belief state [Astrom 1965]. For clarity, we have redone the entire proof from scratch (and restated the theorem to a clearer form), now including detailed intermediate steps, and not building upon prior work (c.f. Appendix D). Note that the equation in our theorem is indeed an equality. The scaling factor introduced in Sutton’s policy gradient theorem is only needed when summing over all states using a discounted state distribution. When summing over the timesteps in an episode with a fixed horizon, one does not need this scaling factor.

---

> > > ### Author Response · Authors · 2024-11-21
> > > **Rebuttal (part 3)**
> > >
> > > > *Line 370: This experiment begins from a specific initialization point. The maximum expected reward is slightly lower than the cooperation reward of 1. Although this is a challenging case, what would results look like for different initialization points? Can a learning-aware agent achieve better outcomes (with an expected reward greater than 1)?*
> > >
> > > At the end of the naive learner’s learning trajectory, the fixed meta agent’s policy will have shaped it into unfair cooperation, resulting in a reward higher than 1. However, the expected reward that we show in Figure 3A is the average reward over the whole learning trajectory of the naive learner against the considered meta agent’s policy. As the naive learner starts from a random policy, it will often defect in the early inner episodes, thereby lowering the average reward for the meta agent. In Figure 3A of our initial submission, we used 10 inner episodes, which resulted in an average reward below 1 (even though the reward at the 10th inner episode is above 1). We remade Figure 3 now with 50 inner episodes, clearly showing that the meta agent extorts the naive learner into unfair cooperation, and thereby achieving an average reward of greater than 1. Initializing the meta agent with a random policy instead of unconditional defection made it faster to learn an extortion policy, however, the final policies upon convergence are similar to those starting from an unconditional defection initialization (data not shown).
> > >
> > > In Appendix C.1, we now include additional results on the analytical iterated prisoner’s dilemma, showing that the extortion policies learned by the meta agent are closely related to the Zero-Determinant extortion strategies first discovered by Press & Dyson 2012. Furthermore, we show that mutual defection is not a Nash Equilibrium anymore in the mixed group setting of Figure 3C, and provide further insights on the parameter learning trajectory of the meta agents, showing that playing against naive learners is a key component for finding cooperative solutions among multiple meta agents (c.f. Appendix C.1).
> > >
> > > > *Figure 3B and 3C: Does the proposed method require interaction with naive agents to initiate training?*
> > >
> > > Figure 3B and 3C both consider a mixed group setting, where a meta agent plays both against another meta agent, as well as randomly initialized naive learners. In Figure 3B, both meta agents first only play against naive learners (the gray shaded region in the plot). As in this setting, the meta agents learn a pure exploitation strategy, their average reward against naive learners is high (blue curves), whereas their average reward when evaluated against the other meta agent is low (red curves) because they both try to extort the other agent. Then after 1500 training steps, we switch the training regime and let the meta agents only play against the other meta agent (non-shaded region in the plot). As both meta agents try to extort the other meta agent into unfair cooperation, and update their parameters based on the resulting trajectories, they will both influence the other to cooperate more, ultimately leading to mutual cooperation.
> > >
> > > In Figure 3C, we let the meta agents play against a mix of other meta agents and naive learners: with probability p_naive, they play against a naive learner, and with probability (1-p_naive), they play against the other meta agent. In the last figure on the right, we let the meta agents only play against other meta agents, without involving any naive learners.
> > >
> > > > *Why is there no comparison between COALA and M-FOS in Figure 5?*
> > >
> > > Figure 5 shows the training curves of both COALA agents and M-FOS agents, when playing against agents of the same type (COALA or M-FOS resp.) and naive learners. We are currently implementing a round-robin setting where we can let COALA agents interact with M-FOS agents and other baselines during training, and will include these results in the final version of the paper.
> > >
> > > > *Why are all references in blue font? This formatting seems differ from other submissions.*
> > >
> > > Fixed.
> > >
> > > > *In Lines 14, 33, 49, 106, 122, 161, 185, 478, and 479, the single quotation marks should be replaced with double quotation marks.*
> > >
> > > Fixed.

---

> > > > ### Author Response · Authors · 2024-11-25
> > > > **Gentle reminder**
> > > >
> > > > Gentle reminder: The discussion period closes in 2 days. We've tried to address all your concerns with new results, clarifications and an updated manuscript. Please let us know if you have any remaining concerns. We look forward to a productive discussion.

---

> > > > > ### Comment · Reviewer_UL6h · 2024-11-27
> > > > >
> > > > > Thank you for the detailed response. However, I still have some concerns, and I intend to maintain my current score.
> > > > >
> > > > > > Me: The metric used in this work is ambiguous. Could you clarify which metric is applied in subsequent sections?
> > > > > > Authors: We now clearly state in the beginning of Section 5 that our aims are twofold: to understand (i) when meta-agents succeed in exploiting naive agents, and (ii) when cooperation is achieved among meta-agents.
> > > > >
> > > > > This paper seems to aim at proposing a very general framework, and the issue of the metrics discussed is critical because they might serve as a guiding principle for the MARL community in the future. Intuitively, I currently believe that the choice of these two metrics is reasonable and self-consistent. However, I think these metrics are not proportionate to the contribution expected from such a general framework.
> > > > >
> > > > > Specifically, "exploiting naive agents" seems like a rather marginal point. Furthermore, if future research focuses solely on outperforming this metric, it could lead to unhealthy trends in the MARL community.
> > > > >
> > > > > On the other hand, "cooperation is achieved among meta-agents" is a better metric than the previous one. However, the authors' conclusions seem to have been derived from the experimental results, rather than being reflected in their methods. **This metric does not appear in the optimization objective of the learning agents but instead emerges as a result.** This is why I previously questioned whether this is an **agent-based simulation**. For instance, why would optimizing the meta-agent’s own expected rewards, according to the authors’ method, naturally lead to outcomes that are more Pareto-dominant?
> > > > >
> > > > > I believe the authors should include more discussion on why these two metrics were chosen, rather than merely revising the paper according to my suggestions or based on experimental outcomes.
> > > > >
> > > > > ---
> > > > >
> > > > > > Me: Using sampling method does not seem to solve the non-stationarity issue. And this contradicts the motivation of this method.
> > > > > > Authors: the sequence model meta-learns to infer “in-context” the parameters and learning algorithms employed by co-players based solely on observations including their past actions.
> > > > >
> > > > > Now I am certain that the authors' method does not explicitly input the policy parameters of others into their own network. However, the authors’ response highlights the exact issue I am concerned about. The meta-agent takes into account the historical actions of others, but these different historical actions are sampled at different stages of policy updates. For example, $a_1$ is sampled according to $\pi_0$, then $\pi_0$ is updated to $\pi_1$, and $a_2$ is sampled accordingly.
> > > > >
> > > > > If the authors claim that their estimated gradients are unbiased, which update stage of the policy is being estimated? Additionally, is the volume of sampled data sufficient for the learning algorithm to estimate the co-player’s policy parameters and use them to update its own policy?
> > > > >
> > > > > ---
> > > > >
> > > > > PS: The updated PDF has become significantly larger, nearly twelve times the size of the previous version.

---

> > > > > > ### Author Response · Authors · 2024-11-27
> > > > > > **Response (Part 1)**
> > > > > >
> > > > > > We thank the reviewer for taking the time to read our rebuttal and raising detailed comments. We reply below to each comment.
> > > > > >
> > > > > > > This paper seems to aim at proposing a very general framework, and the issue of the metrics discussed is critical because they might serve as a guiding principle for the MARL community in the future. Intuitively, I currently believe that the choice of these two metrics is reasonable and self-consistent. However, I think these metrics are not proportionate to the contribution expected from such a general framework. [...] However, the authors' conclusions seem to have been derived from the experimental results, rather than being reflected in their methods. This metric does not appear in the optimization objective of the learning agents but instead emerges as a result. This is why I previously questioned whether this is an agent-based simulation. For instance, why would optimizing the meta-agent’s own expected rewards, according to the authors’ method, naturally lead to outcomes that are more Pareto-dominant?
> > > > > >
> > > > > > We thank the reviewer for raising this point, as it goes to the core of one of the main contributions of our work: we show how cooperation among meta agents can arise from optimizing uncoordinated, purely selfish objectives.
> > > > > >
> > > > > > It can help to situate our work in the broader multi-agent RL (MARL) field. There are two main approaches in the mixed-motive MARL literature to solve mixed-motive games: (i) learning-aware RL methods (such as LOLA, M-FOS, …) and (ii) reward-shaping to change the incentive structure of the game, making cooperative equilibria easier to find [Hughes et al. 2018; Jaques et al. 2019; Du et al. 2023]. Our work is situated in the former line of work, as we propose a new learning-aware policy gradient algorithm that is compatible with any general-sum reward structure. Hence, **including the social welfare metric in the optimization objective of the learning agents would defeat the purpose of our approach in showing that cooperation can be achieved with purely selfish objectives**. Furthermore, including the social welfare inside the objective of the learning agents changes the problem from a social dilemma general sum game to a pure collaborative setting, where agents intrinsically care about the rewards of other agents.
> > > > > >
> > > > > > Our new MARL method consists out of two parts: (i) we introduce COALA-PG, an improved policy gradient method to shape naive learners (c.f. Section 3) , and (ii), a novel population-based MARL training approach where we train meta agents against a mix of naive learners and other meta agents, showing that in this novel population-based training setup, meta-agents achieve cooperative equilibria which are Pareto dominant to the equilibria achieved by only training against other meta agents, as shown in Section 4. We emphasize here the novel results presented in this section:  despite the fact that the agents have purely selfish objectives,  they still converge to a cooperative solution. We show in Section 4 the mechanisms at the core of this interesting property of our new method.
> > > > > >
> > > > > > “Exploiting naive agents” is indeed only an instrumental goal towards the main goal of achieving cooperation among meta agents. Yet, this co-player shaping component of our approach is crucial for achieving cooperation among meta agents. Hence, a main part of our paper should validate our new co-player shaping method COALA-PG, against other state-of-the-art co-player shaping methods in the field, which we do in figures 5 and 6. We agree with the reviewer that “cooperation among meta agents” is the main metric of interest, and we note that our work guides the field of co-player shaping towards this ultimate goal, as many previous state-of-the-art methods solely focused on “exploiting naive agents” [Khan et al. 2024].
> > > > > >
> > > > > > Similar to agent-based modeling, we indeed simulate agents equipped with our methods in a variety of environments in Section 5 to validate our approach, which is standard practice in the MARL field. We highlight however that we do not just quantify the observed behavior of the agents. We reiterate that, in Section 4, we uncover the main mechanisms arising from co-player shaping which we then leverage to achieve cooperation among meta agents by designing a **new population-based training method where meta agents train against a mixed group of naive learners and other meta agents**. The reviewer’s question “why would optimizing the meta-agent’s own expected rewards, according to the authors’ method, naturally lead to outcomes that are more Pareto-dominant?” is at the heart of the core contribution of our work, and we dedicate the whole of Section 4, combined with additional analysis in Appendix C.1 to answering this question.
> > > > > >
> > > > > > We hope the above clarifications address the reviewer’s remaining concerns about the used metrics, and remain open to discuss further concerns. We will clarify the above mentioned main contributions in the final version of our paper.

---

> > > > > > > ### Author Response · Authors · 2024-11-27
> > > > > > > **Response (part 2)**
> > > > > > >
> > > > > > > > Now I am certain that the authors' method does not explicitly input the policy parameters of others into their own network. However, the authors’ response highlights the exact issue I am concerned about. The meta-agent takes into account the historical actions of others, but these different historical actions are sampled at different stages of policy updates. For example, $a_1$  is sampled according to $\pi_0$, then $\pi_0$ is updated to $\pi_1$, and $a_2$ is sampled accordingly. If the authors claim that their estimated gradients are unbiased, which update stage of the policy is being estimated? Additionally, is the volume of sampled data sufficient for the learning algorithm to estimate the co-player’s policy parameters and use them to update its own policy?
> > > > > > >
> > > > > > > The meta agent’s policy is trained by our policy gradient method to optimally shape the naive co-player. As a part of this optimal policy, the sequence model meta-learns to infer “in-context” the parameters and learning algorithms employed by co-players. How exactly the sequence model performs this inference is a black box, but one possible approach would be a Kalman-filter-like method. In such approach, the sequence model combines previous actions arising from a series of different co-player policies (as they are updated along the trajectory, as the reviewer correctly notes), together with an implicitly learned model of how co-player policies evolve over meta trajectories, into a best estimate of the **current co-player’s parameters**. In contrast to e.g. fictitious play which uses previous actions to estimate an “average co-player policy” based on empirical frequencies, an optimal meta-policy optimally integrates previous actions to infer the current co-player’s parameters at each point in time, to optimally inform which action to take to shape the future behavior of the co-player.
> > > > > > >
> > > > > > > The sequence policy at timestep $l$ estimates the current co-player’s policy at timestep $l$, hence it does not need to “choose” a specific stage of the co-player’s policy. We not only “claim” that our estimated gradients are unbiased, we **prove** it from scratch in Appendix D (and the same result can be obtained by combining our batched co-player POMDP with Sutton’s policy gradient theorem and Astrom’s transformation of a POMDP in a belief state MDP). If the reviewer has remaining doubts on the unbiasedness of COALA-PG, please point us to the relevant sections in the proof such that we can clarify this.
> > > > > > >
> > > > > > > We empirically show in Section 5 that COALA-PG can successfully shape naive learners in a variety of environments, under similar volumes of sampled environment interactions compared to previous state-of-the-art work [Lu et al. 2022, Khan et al. 2024], indicating that indeed, the volume of sampled data is sufficient for estimating the co-player’s policies and update the meta agent’s own parameters.
> > > > > > >
> > > > > > > > PS: The updated PDF has become significantly larger, nearly twelve times the size of the previous version.
> > > > > > >
> > > > > > > We indeed updated the figures to a higher resolution and increased font sizes, as requested by other reviewers.
> > > > > > >
> > > > > > > We hope the above clarifications address the remaining concerns of the reviewer. Please let us know if there are any other remaining concerns preventing the reviewer from raising their score.

---

> > > > > > > > ### Comment · Reviewer_UL6h · 2024-12-02
> > > > > > > >
> > > > > > > > Thanks for the detailed response; I have adjusted my score accordingly.

---

### Official Review · Reviewer_MZki · 2024-11-03

**Soundness:** 2
**Presentation:** 4
**Contribution:** 3
**Rating:** 8
**Confidence:** 4

**Summary:**

The paper brings sound improvements over the current opponent shaping methods for non-cooperative general-sum games with self-interested learners. The core contributions over the existing methods are introducing a policy-gradient method (COALA-PG) capable of shaping other learners without making any assumptions on learners' optimization functions due to the sequential and batched nature of the expected return to be maximized. The method is evaluated on Iterated Prisoner's Dilemma and CleanUp environments. COALA-PG is compared to M-FOS and variants of itself as a baseline and investigation of how the pool of learners, whether they are meta or naive learners, affect reaching cooperation.

**Strengths:**

* Introduction of a novel unbiased policy gradient formulation to shape naive learners that does not need to know the learners' optimization functions but can infer them by sequence modeling.
* Comparisons showing the effect of how learners are pooled (meta vs naive or meta vs meta etc.) give insight into how shaping is achieved.
* COALA-PG improves performance over M-FOS.
* The figures are informative.

**Weaknesses:**

* There is insufficient explanation on why COALA-PG is so much better than batch-unaware COALA-PG which perform very similar to M-FOS. The credit assignment figure is a nice visualization to give intuition, but it does not provide concrete explanation and we have to speculate. Since batch-awareness is the main factor that makes COALA-PG's performance superior, it is difficult to attribute the success to sequence modeling and inferring the learning dynamics. There should be an investigation into how batch-awareness affect and change gradients and why batch-unaware COALA-PG is performs similar to MFOS.
* Baselines are insufficient. The only shaping method compared is the MFOS. Although mentioned in the paper, no comparison to Shaper is done in experiments. There is no comparison to LOLA either; even though MFOS is already expected to outperform LOLA, still a verification should have been done.
* There is no versus for different shaping methods, i.e. there is no experiment on how COALA-PG shapes M-FOS and vice versa.

**Questions:**

* Why is there no experiment for LOLA?
* Why does batch-unaware COALA-PG perform so similar to M-FOS, while the batch-aware COALA-PG surpasses it? Did you investigate how gradients change and why the batch-aware gradients work so well?

---

> ### Author Response · Authors · 2024-11-20
> **Rebuttal (Part 1)**
>
> We thank the reviewer for the constructive comments that has helped us improve our paper. We reply to each point separately below.
> > *There is insufficient explanation on why COALA-PG is so much better than batch-unaware COALA-PG which perform very similar to M-FOS. The credit assignment figure is a nice visualization to give intuition, but it does not provide concrete explanation and we have to speculate. Since batch-awareness is the main factor that makes COALA-PG's performance superior, it is difficult to attribute the success to sequence modeling and inferring the learning dynamics. There should be an investigation into how batch-awareness affect and change gradients and why batch-unaware COALA-PG is performs similar to MFOS.
> [...]
> Why does batch-unaware COALA-PG perform so similar to M-FOS, while the batch-aware COALA-PG surpasses it? Did you investigate how gradients change and why the batch-aware gradients work so well?*
>
> We thank the reviewer for this excellent comment. To understand what is going on, it is helpful to differentiate the contributions to the policy gradient arising from the current inner episode return, and the contributions arising from future inner episode returns. The former provides a learning pressure to play well against the current policy of other agents, whereas the latter provides co-player shaping pressure. The main difference between COALA-PG and M-FOS (Eq. 3 vs Eq. 26) is that COALA-PG scales the return of the current inner episode by 1/B, whereas M-FOS does not. This scaling is crucial, as the the influence of an action on future inner episodes is of order O(1/B) because the naive learner averages its update over the B inner episode trajectories. Hence, by scaling the inner episode return by 1/B, COALA-PG ensures that both contributions (current and future episodes) have the same scaling w.r.t. B, and finally by summing the policy gradient over the minibatch instead of averaging, we end up with a policy gradient of O(1). M-FOS and batch-unaware COALA do not scale the current inner episode return, which causes the co-player shaping learning signal to vanish w.r.t. the current inner episode return, resulting in poor shaping performance. Ultimately, this problem can be traced back to a misspecification of the underlying POMDP, which does not take into account inner episode minibatches. The key message here is that the proposed batched co-player shaping POMDP with appropriate scaling solves the issue.
>
> To clarify this we have implemented three changes in the paper:
> * We now include an additional analysis comparing the contribution of the current and future episodes for the different learning rules considered in the paper. Specifically, figure 12 in appendix H.1 tracks throughout meta-training the contributions to the parameter update that stem from current vs. future episodes, with the latter measuring the shaping influence of an action on co-player learning. Our analysis shows that COALA-PG indeed balances the two contributions, maintaining an O(1) ratio, whereas the other learning rules do not.
> * We have included a new visualization of the difference between the COALA-PG update and the batch-unaware policy gradient update (see Fig. 11 in App. G)
> * We have rewritten Theorem 3.1 and the passage below to discuss this point. It now reads:
> >There are three important differences between `COALA-PG` and naively applying policy gradient methods to individual trajectories in a batch. (i) Each gradient term for an individual action $a_l^{i,b}$ takes into account the future inner episode returns averaged over the whole minibatch, instead of the future return along trajectory $b$ (see Fig. 2). This allows taking into account the influence of this action on the parameter update of the naive learner, which influences all trajectories in the minibatch after that update. (ii) Instead of averaging the policy gradients for each trajectory in the batch, `COALA-PG` accumulates (sums) them. This is important, as otherwise the learning signal would vanish in the limit of large minibatches. Intuitively, when a naive learner uses a large minibatch for its updates, the effect of a single action on the naive learner's update is small ($O(1/B)$), and this must be compensated by summing all such small effects. (iii) To ensure a correct balance between the return from the current inner episode $m_l$ and the return from future inner episodes, `COALA-PG` rescales the current episode return by $1/B$. Figure 12 in App. H shows empirically that `COALA-PG` correctly balances the policy gradient terms arising from the current inner episode return versus the future inner episode returns, whereas M-FOS (Lu et al., 2022) and a naive policy gradient that ignores the other parallel trajectories over-emphasize the current inner episode return, causing them to lose the co-player shaping learning signals.

---

> > ### Author Response · Authors · 2024-11-20
> > **Rebuttal (Part 2)**
> >
> > > *Baselines are insufficient. The only shaping method compared is the MFOS. Although mentioned in the paper, no comparison to Shaper is done in experiments. There is no comparison to LOLA either; even though MFOS is already expected to outperform LOLA, still a verification should have been done.
> > [...]
> > Why is there no experiment for LOLA?*
> >
> > We now include LOLA in our experiments, including a more challenging variant of CleanUp-Lite: CleanUp-lite-2D. First, we find that when equipping LOLA with the state-of-the-art Loaded-DiCE higher-derivative estimator [Farquhar et al. 2019], and searching for good hyperparameters, we can make LOLA cooperate on the iterated prisoner’s dilemma with sequence model policies, in contrast to what previous findings suggest [Zhao et al. 2022]. When scaling to the more complex CleanUp-lite-2D environment, the LOLA baseline is outperformed by our COALA-PG estimator, suggesting that the model-free COALA-PG approach is more suitable to scale to larger environments (see Fig. 13 in App. H.2). We further highlight that COALA-PG does not require higher-order derivatives, nor explicit access to co-player parameters.
> >
> > Furthermore, we include a novel analysis of the emergence of cooperation between agents trained with the LOLA algorithm. Our analysis reveals a crucial role played by a hyperparameter of the algorithm, namely the number of simulated co-player learning steps (often referred to as ``look-ahead steps”). As this number increases, LOLA agents transition from cooperation to extortion strategies resulting in mutual defection. Our analysis explains the origin of this phenomenon, and the similarities and differences between LOLA and COALA.
> >
> > > *There is no versus for different shaping methods, i.e. there is no experiment on how COALA-PG shapes M-FOS and vice versa.*
> >
> > We are currently implementing a round-robin tournament in our codebase and will include the results in the final version of our paper.

---

> > > ### Author Response · Authors · 2024-11-25
> > > **Gentle reminder**
> > >
> > > Gentle reminder: The discussion period closes in 2 days. We've tried to address all your concerns with new results, clarifications and an updated manuscript. Please let us know if you have any remaining concerns. We look forward to a productive discussion.

---

> > > > ### Comment · Reviewer_MZki · 2024-11-27
> > > >
> > > > I thank the authors for all the insightful clarifications, and the new results with the updated manuscript that improved the paper a lot. My concerns have been addressed, and I believe with all the additions in the updated manuscript, the paper is well enough to be published. I am raising my score.

---

> > > > > ### Author Response · Authors · 2024-11-27
> > > > >
> > > > > We thank the reviewer for taking the time to go through our rebuttal and reevaluating their score.

---

### Official Review · Reviewer_fP5Y · 2024-11-03

**Soundness:** 3
**Presentation:** 3
**Contribution:** 3
**Rating:** 6
**Confidence:** 4

**Summary:**

The paper introduces a novel policy gradient algorithm for learning-aware reinforcement learning, which allows agents to model the learning dynamics of other agents. This approach enables cooperative behavior among self-interested agents, even in complex scenarios like temporally extended action coordination. The algorithm avoids higher-order derivatives, and can handle mini-batched learning. Additionally, the paper provides a theoretical explanation for the emergence of cooperation in the iterated prisoner's dilemma based on learning awareness and agent heterogeneity.

**Strengths:**

1. The paper tackles a core issue in multi-agent learning - fostering cooperation among self-interested agents.
2. The introduction of the first unbiased, higher-derivative-free policy gradient algorithm for learning-aware reinforcement learning is a significant contribution.
3. The proposed algorithm appears to outperform existing methods in achieving cooperation.
4.The paper proposes a novel explanation for cooperation emergence based on learning-awareness and agent heterogeneity in the iterated prisoner's dilemma.

**Weaknesses:**

In my view, the “elegant” formulation and the batched co-player shaping POMDP appear similar to a parallel environment POMDP setup, while COALA-PG seems more like a delayed optimization applied after several episodes. This raises concerns about the true novelty and contribution of the work. I have a few specific questions regarding this point.

It’s entirely possible that I may not yet have fully understood the paper. However, improving the paper’s clarity and structure would make it more accessible and easier to follow.

I look forward to discussing these points with the authors, and I am open to raising my score if my concerns are addressed satisfactorily.

**Questions:**

1.	What are the key differences between the proposed batched co-player shaping POMDP and a standard POMDP with parallel environments? Equation 3 appears to simply sum rewards across parallel environments, making it unclear what distinct advantages this formulation provides.
2.	The font size in the figures is too small, making it difficult to read. Enhancing font size and clarity would significantly improve readability.
3.	 There are well-known baseline methods in the mixed-motive game literature, particularly within the Cleanup environment. Why does the paper not include comparisons with these established methods?
4.       More information is needed on the simplified Cleanup environment. How does it differ from the standard Cleanup environment, and what are the implications of these differences for the study’s findings?
5.	 While the paper aims to address a mixed-motive game, the problem is formulated as a general-sum game. In my view, general-sum games are not fully equivalent to mixed-motive games. Could you clarify the rationale behind this formulation choice?

---

> ### Author Response · Authors · 2024-11-20
> **Rebuttal (Part 1)**
>
> We thank the reviewer for the constructive comments that have helped us improve the clarity of our paper. We reply to each point separately below.
>
> > *In my view, the “elegant” formulation and the batched co-player shaping POMDP appear similar to a parallel environment POMDP setup, while COALA-PG seems more like a delayed optimization applied after several episodes. This raises concerns about the true novelty and contribution of the work. I have a few specific questions regarding this point.*
> > *What are the key differences between the proposed batched co-player shaping POMDP and a standard POMDP with parallel environments? Equation 3 appears to simply sum rewards across parallel environments, making it unclear what distinct advantages this formulation provides.*
>
> This is an important point, and we actually did attend to this matter with our controls, so we clarified it in the updated manuscript. In the manuscript, we referred to standard POMDPs with parallel environments as “batch-unaware”, because the parallel trajectories do not interact with each other. In contrast, in our batched co-player shaping POMDP, the parallel trajectories interact with each other through the parameter updates of the naive learner: after each inner episode, the naive learners update their parameters based on all the parallel current inner episodes. Hence, actions of the meta agent in parallel trajectory $b$ can influence the naive agent’s parameters used in other parallel trajectories $b’\neq b$. As a consequence, the COALA policy gradient (Eq. 3) averages future returns over all parallel batches after a specific action $a_l^{i,b}$, instead of only considering the future return of trajectory $b$ which would be the case in the batch-unaware POMDP (i.e. parallel environments).
>
> The standard parallel environment POMDP leads to an update that undervalues the effect of an action on future-episode behavior of co-players, as it only includes a single trajectory in its calculation of the updates. In other words, it overemphasizes current-episode performance over shaping co-player learning. This results in poor performance of the batch-unaware COALA baseline on all experiments considered (c.f. Section 5), compared to our COALA-PG method that correctly handles the batched co-player POMDP, thereby further illustrating that our contribution of correctly formulating the batched co-player shaping POMDP is crucial for achieving good performance.
> To clarify this we have implemented three changes in the paper:
> * We now include a new experimental analysis measuring the present-episode vs. future-shaping contributions to the gradient for different estimators throughout meta-training, see Fig. 12 in Appendix H.1. COALA-PG balances the two, achieving a ratio of order one, whereas standard (batch-unaware) learning and M-FOS do not.
> * We have rewritten Theorem 3.1 and the passage below to discuss this point. It now reads:
> >There are three important differences between `COALA-PG` and naively applying policy gradient methods to individual trajectories in a batch. (i) Each gradient term for an individual action $a_l^{i,b}$ takes into account the future inner episode returns averaged over the whole minibatch, instead of the future return along trajectory $b$ (see Fig. 2). This allows taking into account the influence of this action on the parameter update of the naive learner, which influences all trajectories in the minibatch after that update. (ii) Instead of averaging the policy gradients for each trajectory in the batch, `COALA-PG` accumulates (sums) them. This is important, as otherwise the learning signal would vanish in the limit of large minibatches. Intuitively, when a naive learner uses a large minibatch for its updates, the effect of a single action on the naive learner's update is small ($O(1/B)$), and this must be compensated by summing all such small effects. (iii) To ensure a correct balance between the return from the current inner episode $m_l$ and the return from future inner episodes, `COALA-PG` rescales the current episode return by $1/B$. Figure 12 in App. H shows empirically that `COALA-PG` correctly balances the policy gradient terms arising from the current inner episode return versus the future inner episode returns, whereas M-FOS (Lu et al., 2022) and a naive policy gradient that ignores the other parallel trajectories over-emphasize the current inner episode return, causing them to lose the co-player shaping learning signals.
> * We include a new visualization of the difference between the batched co-player shaping POMDP and the batch-unaware POMDP (Fig. 11 in Appendix G).
> We hope that these changes and additional explanations make the novelty and contribution of our work more clear. We remain open to address any further questions the reviewer might have on this important point.

---

> > ### Author Response · Authors · 2024-11-20
> > **Rebuttal (Part 2)**
> >
> > > *The font size in the figures is too small, making it difficult to read. Enhancing font size and clarity would significantly improve readability.*
> >
> > Fixed. We have now increased the font size for Figs. 3, 4, 5, 6 and 7.
> >
> > > *There are well-known baseline methods in the mixed-motive game literature, particularly within the Cleanup environment. Why does the paper not include comparisons with these established methods?*
> >
> > There are two main approaches in the mixed-motive multi-agent RL (MARL) literature to solve mixed-motive games: (i) learning-aware RL methods (such as LOLA, M-FOS, …) and (ii) reward-shaping to change the incentive structure of the game, making cooperative equilibria easier to find [Hughes et al. 2018; Jaques et al. 2019; Du et al. 2023]. Our work is situated in the former line of work, as we propose a new learning-aware policy gradient algorithm that is compatible with any general-sum reward structure. Hence, the baselines we compare to are also situated in the learning-aware MARL methods.
> >
> > Besides the M-FOS method, we include a new baseline in our results: LOLA. LOLA is an important and established learning-aware MARL method. First, we find that when equipping LOLA with the state-of-the-art Loaded-DiCE higher-derivative estimator [Farquhar et al. 2019], and searching for good hyperparameters, we can make LOLA cooperate on the iterated prisoner’s dilemma with sequence model policies, in contrast to what previous findings suggest [Zhao et al. 2022]. When scaling to the more complex CleanUp-lite-2D environment, the LOLA baseline is outperformed by our COALA-PG estimator, suggesting that the model-free COALA-PG approach is more suitable to scale to larger environments (see Fig. 13 in App. H.2).
> >
> > Furthermore, we include a novel analysis of the emergence of cooperation between agents trained with the LOLA algorithm. Our analysis reveals a crucial role played by a hyperparameter of the algorithm, namely the number of simulated co-player learning steps (often referred to as ``look-ahead steps”). As this number increases, LOLA agents transition from cooperation to defection. Our analysis explains the origin of this phenomenon, and the similarities and differences between LOLA and COALA.
> >
> > The CleanUp environment has mainly been investigated within the reward-shaping line of work, relying on modifying the objectives of each agent, explicitly rewarding pro-social behavior. While this is an interesting line of work on its own, it is out of scope for the current paper, which focuses on the classical (more challenging) setting of purely self-interested agents. We further note that other related learning-aware MARL methods in the literature [Lu et al. 2022; Khan et al. 2024; Aghajohari et al. 2024] also only compare to other learning-aware MARL methods, and not to reward-shaping MARL methods.
> >
> > > *While the paper aims to address a mixed-motive game, the problem is formulated as a general-sum game. In my view, general-sum games are not fully equivalent to mixed-motive games. Could you clarify the rationale behind this formulation choice?*
> >
> > We consider mixed-motive games as general-sum games that are neither zero-sum, nor fully-cooperative (i.e., where all agents share the same objective). We have now clarified in section 2 that we focus on this particular type of general-sum game, and provide our motivation for doing so. It reads as follows:
> >
> > We focus on general-sum games, where each agent has their own reward function, possibly different from those of other agents. Specifically, we consider mixed-motive general-sum games that are neither zero-sum nor fully-cooperative. Analyzing and solving such general-sum games while letting every agent individually and independently maximize their rewards [a setting often referred to as ``fully-decentralized reinforcement learning''; Albrecht et al. 2024] is a longstanding problem in the fields of machine learning and game theory for two primary reasons, described below.

---

> > > ### Author Response · Authors · 2024-11-20
> > > **Rebuttal (Part 3)**
> > >
> > > > More information is needed on the simplified Cleanup environment. How does it differ from the standard Cleanup environment, and what are the implications of these differences for the study’s findings?
> > >
> > > We have now expanded the discussion of the environment on Appendix B.1. Note that we included new experiments on a 2D version of our Cleanup-Lite environment (see Appendix H.2). The description of the environments read:
> > > ### CleanUp-lite
> > >
> > > `CleanUp-lite` is a simplified two-player version of the `CleanUp` game, which is part of the Melting Pot suite of multi-agent environments [1]. It is modelled as follows:
> > >
> > > *   **State:** The world is a grid of size 3, arranged horizontally. The right extremity is the river, and the left one the orchard. The world state also contains the pollution level of the river  $P$, number of apples currently available in the orchard $A$, the position of each agent, and their respective zapped state.
> > > *   **Action:** There exist 4 actions: {move right, move left, zap, do nothing}.
> > > *   **Dynamics:** The environment evolves at every timestep in the following order:
> > >     1.  The pollution level increases by 1 with a fixed probability $p_\text{pollution}=0.5$, and capped at $P_\text{max}=5$.
> > >     2.  The apple grows by 1 with probability $1-\min(1, P/P_\text{threshold})$, where $P_\text{threshold}=3$, and capped at $A_\text{max}=4$.
> > >     3.  Then, an agent in the "orchard" cell who is not zapped, has a probability $A/A_\text{max}$ of picking up an apple, independently from one another, in which case the apple level decreases by 1 (capped at 0).
> > >     4.  An agent in the "river" cell who is not zapped has a probability $p_\text{clean}=0.7$ of decreasing the pollution level by 1, independently of other agents (capped at 0).
> > >     5.  Finally, an agent zapping has a $p_\text{zap}=0.8$ probability of successfully zapping the opponent who is in the same cell. If the zapping is successful, the opponent is frozen for $t_\text{zap}=3$ timesteps, during which it is frozen and cannot be further zapped.
> > > *   **Initial state:** Agents are randomly placed on the grid, unzapped, $A$ is set to $0$, $P$ to $P_\text{threshold}$.
> > > *   **Observation:** The observation is fully observable. Each agent sees the position of each agent encoded as concatenated one-hot vectors indicating the position in the grid, the normalized number of apples $A/A_\text{max}$, normalized pollution level $P/P_\text{max}$, the normalized remaining timeout status (zapped or not), and if so, a countdown indicating how many timesteps remain for each agent ($t_\text{agent}/t_\text{zap}$ where $t_\text{agent}$ is the time left before the agent unfreezes). The observation is symmetric.
> > > *   **Reward:** An agent that picks up an apple receives a reward of $r_\text{apple}=1$ in that timestep.
> > >
> > > > *It’s entirely possible that I may not yet have fully understood the paper. However, improving the paper’s clarity and structure would make it more accessible and easier to follow.*
> > >
> > > We hope the above changes and clarifications are sufficient, but we remain open to answer any further questions you may have.

---

> > > > ### Author Response · Authors · 2024-11-25
> > > > **Gentle reminder**
> > > >
> > > > Gentle reminder: The discussion period closes in 2 days. We've tried to address all your concerns with new results, clarifications and an updated manuscript. Please let us know if you have any remaining concerns. We look forward to a productive discussion.

---

> > > ### Comment · Reviewer_fP5Y · 2024-11-26
> > >
> > > Although the paper is not a reward-shaping method, I am still curious about how its performance compares to that of reward-shaping approaches. Could it outperform reward-shaping-based methods? If not, what insights can be drawn from this? Including such insights in a future version could further enhance the paper.

---

> > > > ### Author Response · Authors · 2024-11-27
> > > >
> > > > We thank the reviewer for taking the time to go through our rebuttal, raising their score, and engaging in the discussion.
> > > >
> > > > To respond to the reviewer’s last suggestion, we would like to expand on the rationale we followed for not having compared directly to pro-social/reward shaping methods.
> > > >
> > > > Reward shaping, typically done by adding (part of) the other agent’s rewards to each agent’s individual objective, has two main effects on multi-agent learning. First, it changes the environment by removing conflicting interests. For example, when taking the total sum of rewards of all agents as individual objectives, the prisoner’s dilemma ceases to be a social dilemma. Second, it makes credit assignment more challenging for the individual agents, as they do not know whether the received reward originated from itself or other agents. Hence, in multi-agent games where the main challenge is conflicting interests, and credit assignment is not a significant issue (such as the iterated prisoner’s dilemma), reward shaping methods that remove conflicting interests will likely perform on par or better than COALA-PG. In games where credit assignment is a significant part of the challenge, reward shaping methods would significantly increase the credit assignment challenge, which could possibly outweigh the benefits of removing conflicting interests. This makes it hard to predict which method would come out on top.
> > > >
> > > > The motivation for our work, and for investigating learning awareness more generally, is fundamentally different from the reward shaping field: the aim is to design novel multi-agent RL algorithms compatible with a wide range of general sum games, including social dilemmas, instead of changing the environment into a limited range of general sum games, excluding social dilemmas, that are compatible with out-of-the-box single-agent RL algorithms. Apart from the usefulness of discovering fundamentally new MARL algorithms, the learning-awareness approach has the advantage of not requiring specific environment settings that are not always present in real-world settings. (i) reward-shaping methods need privileged access to other agents’ rewards, which is not possible in decentralized training setups. (ii) All players need to agree to change their reward function; if not, nonconforming players can easily exploit the pro-sociality of other agents by selfishly optimizing only their own rewards. This makes the reward-shaping approach less robust to real-world settings with possibly adversarial players.
> > > >
> > > > The main contribution of our work is to show when and how optimizing purely selfish objective functions can give rise to cooperative behavior, by leveraging learning awareness. Hence, even though it would indeed be interesting to see how reward-shaping methods perform, we would not be comparing apples to apples, as (i) reward shaping methods do not optimize purely selfish objectives, but pro-social objectives instead, and (ii) there exist relevant settings where reward shaping methods are not applicable, whereas learning-aware methods are. Consequently, we encourage future research to address the careful comparison of reward-shaping methods and learning-awareness methods, and argue that such comparison would not fit well in our current work.
> > > >
> > > > We hope this clarifies why we did not include a comparison with reward-shaping methods in our work. Please let us know if there are any remaining concerns preventing the reviewer from further raising their score. We look forward to continuing this productive discussion.

---

> ### Comment · Reviewer_fP5Y · 2024-11-26
>
> Thanks for your response.  Your response have solved most my questions.  Thus, I will increase my score.

---

### Official Review · Reviewer_pFwY · 2024-11-04

**Soundness:** 3
**Presentation:** 4
**Contribution:** 3
**Rating:** 8
**Confidence:** 2

**Summary:**

To solve two challenges of general-sum games: 1. Non-stationarity of the environment, hard to choose a good equilibrium. The author propose an unbiased, higher derivative-free policy gradient algorithm for learning-aware RL, which considers how the action will influence the other agent's policy changing: They do policy gradient in meta-episode. They have extensive theoratic comparison of their meta policy gradient with prior works. And have interesting findings in the iterated prisoner's dilemma and a grid-world-like social dilemma "CleanUp".

**Strengths:**

1. Clear writing, clear concept definition.
2. extensive theoratic comparison with prior works and experiments on two non-trivial settings.
3. enough details of implementation in the appendix.

**Weaknesses:**

This may be difficult, but it would be great if you could show the efficiency of your method on more difficult environments of deep MARL beyond matrix game or grid world (like Agar.io[1])

[1]: Discovering Diverse Multi-Agent Strategic Behavior via Reward Randomization, ICLR 2021

**Questions:**

None

---

> ### Author Response · Authors · 2024-11-20
> **Rebuttal**
>
> We thank the reviewer for the encouraging comments.
>
> > *This may be difficult, but it would be great if you could show the efficiency of your method on more difficult environments of deep MARL beyond matrix game or grid world (like Agar.io[1])*
>
> While still being a grid world environment, we note that we have now included a more complicated 2D version of CleanUp-Lite in the paper (see Appendix H.2 in the updated pdf). Our previous findings hold in this more challenging setting. Scaling up learning-aware MARL methods beyond grid worlds is a general unsolved challenge in the field, which we (and other labs) aim to address in future work in the coming years.
>
> We remain open to any other remarks to guide the improvement of our paper that can be addressed within the timeframe of the ICLR rebuttals.

---

> > ### Author Response · Authors · 2024-11-25
> > **Gentle reminder**
> >
> > Gentle reminder: The discussion period closes in 2 days. We've tried to address all your concerns with new results, clarifications and an updated manuscript. Please let us know if you have any remaining concerns. We look forward to a productive discussion.

---

> > > ### Comment · Reviewer_pFwY · 2024-11-26
> > >
> > > Thank you for your reply. I decide not to change my score.

---

### Official Review · Reviewer_GXnN · 2024-11-04

**Soundness:** 2
**Presentation:** 2
**Contribution:** 2
**Rating:** 5
**Confidence:** 4

**Summary:**

This study presents a framework for multi-agent cooperation in partially observable general-sum games to enable agents to adaptively cooperate in non-stationary environments by conditioning their policies on past interactions. The COALA-PG method introduces improved policy calculation through accumulated gradients and variance reduction that enhances stability and computational efficiency. The framework is tested on benchmark tasks such as the Iterated Prisoner´s Dilemma and the Cleanup game, demonstrating cooperative behavior among decentralized agents.

**Strengths:**

1. **History-Dependent Adaptation for Multi-Agent Cooperation:** The paper introduces a promising approach that enables agents to adaptively cooperate by conditioning policy updates on observation histories. This allows agents to respond to non-stationarity in general-sum games, specifically handling the evolving distributions of co-agent strategies as each agent independently learns and adapts over time. By incorporating these historical observations, the framework aims to maintain effective cooperation even as the behaviors of other agents shift dynamically, without explicitly modeling the non-stationarity or learning dynamics of other agents.
2. **Theoretical and Practical Advancements in Policy Gradient Estimation:** Theorems 3.1 and 3.2 present a robust policy gradient estimation method specifically designed for cooperative multi-agent learning. COALA-PG´s use of accumulated gradients and minibatch summation approach improves stability and adaptability without requiring higher-order gradients. This makes COALA-PG more scalable in comparison to methods like LOLA-DICE. Additionally, the use of value function baselines in gradient estimation reduces variance which increases COALA-PG´s practical applicability.
3. **Supporting Results Against Naive Learner Baselines:** Empirical comparisons demonstrate that COALA-PG outperforms naive, non-history-conditioned learners, demonstrating the benefit of history dependency in non-stationary environments.

**Weaknesses:**

1. **Full History Dependency:** While history dependency enables adaptability, it may approximate full observability, especially in discrete environments. By accumulating state-action information over time, agents in discrete settings could essentially reconstruct the environment as if fully observable, reducing the framework´s applicability in scenarios where true partial observability is intended.
2. **Simplistic Experimental Scenarios:** The chosen experiments, such as the Iterated Prisoner’s Dilemma (IPD) and Cleanup, are relatively simple and do not fully demonstrate the framework´s ability to manage partial observability or complex, dynamic cooperation. IPD lacks multi-step state evolution, and while Cleanup provides more interaction, it still operates in a low-dimensional grid setting, which is easier for agents to handle than high-dimensional or real-world environments.
3. **Lack of Convergence Analysis:** COALA-PG lacks formal guarantees of stability or convergence in non-stationary environments. This leaves open questions about COALA-PG´s ability to adapt as agent strategies evolve. The study would benefit from additional theoretical or empirical validation to support its effectiveness in highly dynamic settings.

**Questions:**

1. Given that full history dependency may reduce partially observable settings to fully observable ones in discrete environments, have the authors conducted a sample complexity analysis for handling true partial observability, where agents have limited access to or storage for historical data?
2. Could the authors elaborate on their choice of the Iterated Prisoner´s Dilemma and Cleanup as experimental benchmarks?
3. Could the authors discuss whether any convergence properties have been considered for COALA-PG? Given the absence of a formal convergence analysis, how do the authors anticipate the framework will behave in non-stationary environments?

---

> ### Author Response · Authors · 2024-11-21
> **Rebuttal (Part 1)**
>
> We thank the reviewer for the constructive comments that have helped us improve the clarity of our paper. We reply to each point separately below.
> > *Full History Dependency: While history dependency enables adaptability, it may approximate full observability, especially in discrete environments. By accumulating state-action information over time, agents in discrete settings could essentially reconstruct the environment as if fully observable, reducing the framework´s applicability in scenarios where true partial observability is intended.*
> > *Given that full history dependency may reduce partially observable settings to fully observable ones in discrete environments, have the authors conducted a sample complexity analysis for handling true partial observability, where agents have limited access to or storage for historical data?*
>
> In our batched co-player shaping POMDP, the hidden state does not only contain the environment state, but also the policy parameters and update rules of all naive co-players in the environment. Hence, even in simple discrete environments, our setup is a challenging partially observable setting, as we use modern sequence models as policies for all involved agents. It follows that our results from Section 5 provide empirical support that our COALA framework is applicable in scenarios with true partial observability.
>
> To further address the reviewer’s concerns about the applicability of COALA to partially observable scenarios, we included a new proof of Theorem 3.1, showing from scratch that the COALA policy gradient is a correct policy gradient, even when the environment is not fully observable (see Appendix D).
>
> Finally, we now include a more challenging version of the CleanUp-lite environment, extending it to a larger 2D grid, making it a harder partially-observable setting. More information on this point follows in the next section.
>
> While we agree with the reviewer that a sample complexity analysis would be interesting, we argue that our theoretical results combined with the empirical verifications that our COALA-PG method works in true partially observable environments, are sufficient contributions to introduce our novel framework, and defer more theoretical work on sample complexity to future work.
>
> > *Simplistic Experimental Scenarios: The chosen experiments, such as the Iterated Prisoner’s Dilemma (IPD) and Cleanup, are relatively simple and do not fully demonstrate the framework´s ability to manage partial observability or complex, dynamic cooperation. IPD lacks multi-step state evolution, and while Cleanup provides more interaction, it still operates in a low-dimensional grid setting, which is easier for agents to handle than high-dimensional or real-world environments.*
> > *Could the authors elaborate on their choice of the Iterated Prisoner´s Dilemma and Cleanup as experimental benchmarks?*
>
> Despite its simplicity, the iterated prisoner’s dilemma is still a challenging environment to reach cooperative equilibria, when agents use complex sequence models instead of simple tabular policies. Figure 5 shows that indeed, the iterated prisoner’s dilemma exhibits meaningful performance differences between our COALA-PG method and state-of-the-art baseline methods.
> CleanUp-lite introduces two new challenges: (i) cooperation or defection are now temporally extended action sequences instead of atomic actions, and (ii) the social dilemma becomes more challenging, combining a tragedy of the commons challenge (cleaning vs harvesting) with an iterated prisoner’s dilemma (zapping other agents). This allows us to see further differences between our COALA-PG method and state-of-the-art baselines.
>
> We agree with the reviewer that scaling learning-aware multi-agent RL methods towards high-dimensional settings is an important current challenge of the field. We emphasize that in the current status of the field, we can already observe meaningful differences between different approaches on simple environments, providing us with useful insights on which approaches could scale well to more complicated environments.
>
> To address the reviewer’s concern that our chosen environments might be too simplistic, we included results on a new environment: CleanUp-lite-2D. CleanUp-lite-2D increases the difficulty of our simplified CleanUp environment by turning it into a 2D grid world (see App. B.1 for the environment details). We show that COALA-PG finds high-rewarding cooperative solutions similar to those in the more simple CleanUp-lite environment, and outperform M-FOS and LOLA which fail to cooperate (see App. H2).

---

> > ### Author Response · Authors · 2024-11-21
> > **Rebuttal (Part 2)**
> >
> > > *Lack of Convergence Analysis: COALA-PG lacks formal guarantees of stability or convergence in non-stationary environments. This leaves open questions about COALA-PG´s ability to adapt as agent strategies evolve. The study would benefit from additional theoretical or empirical validation to support its effectiveness in highly dynamic settings.*
> > > *Could the authors discuss whether any convergence properties have been considered for COALA-PG? Given the absence of a formal convergence analysis, how do the authors anticipate the framework will behave in non-stationary environments?*
> >
> > Convergence is notoriously difficult to handle formally for multi-agent systems where agents model each other and is an active and challenging area of research. While we agree with the reviewer that this is a desirable goal, we also argue that meaningful contributions towards multi-agent RL can be made without having such formal guarantees, and a lack of formal guarantees should not be held as a standard to judge relevance of novel work.
> >
> > While we do not have theoretical convergence guarantees for our COALA-PG method in the setting where multiple meta agents play against each other, we do perform a detailed empirical analysis on the learning trajectories of learning-aware agents on the iterated prisoner’s dilemma in Section 4. Here, we show that even though it is a challenging non-stationary setting, we can decompose the learning trajectories of the meta agents into gradient contributions seeking extortion policies that force naive learners into unfair cooperation, and other gradient contributions pushing away from extortion towards mutual cooperation. To the best of our knowledge, this is the first detailed analysis of the convergence of learning-aware policy gradient methods towards cooperative equilibria.
> >
> > Encouraged by the reviewer’s interest in convergence analysis, we included new results that further investigate the policies that the meta agents converge to (see App. C.1). We show that meta-agents playing against naive learners converge to Zero-Determinant extortion strategies (Press & Dyson 2012). Furthermore, we show that unconditional defection is no longer a Nash equilibria in our mixed group training setup, allowing meta agents to escape from defection towards mutual cooperation.
> >
> > Finally, we emphasize that our experiments in Section 5, more specifically Figure 5C and Figure 7, are highly non-stationary, as a group of meta agents synchronously update their parameters, continuously changing the resulting environment for each other. Hence, these results support our claims that COALA-PG is well-adapted to highly non-stationary settings.

---

> > > ### Author Response · Authors · 2024-11-25
> > > **Gentle reminder**
> > >
> > > Gentle reminder: The discussion period closes in 2 days. We've tried to address all your concerns with new results, clarifications and an updated manuscript. Please let us know if you have any remaining concerns. We look forward to a productive discussion.

---

### Official Review · Reviewer_xxyH · 2024-11-04

**Soundness:** 3
**Presentation:** 3
**Contribution:** 2
**Rating:** 6
**Confidence:** 4

**Summary:**

This paper introduces a new learning-aware reinforcement learning rule derived as a policy gradient estimator. Instead of using long histories of samples, the policy gradients consider the stochastic minibatch updates used by other agents. The results show improvements in some mixed environments. Overall, the paper is well-written and well-organized. The notations are standard, and the theorems are sound.

**Strengths:**

1. The paper is well-written and well-organized.
2. Theoretical proofs are complete and sound.

**Weaknesses:**

1. The motivation for proposing COALA-PG is unclear. It’s not obvious whether the issue is related to variance or other issues when using mini-batches. The authors suggest that larger mini-batches could pose a problem, but this may lead to higher variance in reward summation. However, these points are not extensively discussed in the manuscript. Additionally, compared to M-FOS, it appears that COALA-PG uses the 1/B term to scale rewards, but it seems that this scaling is still related to controlling the policy gradient’s variance.

2. Why not use a Q-function instead of summing rewards?

3. In Theorem 3.1, what is the advantage of using batches of inner episode histories instead of a long history? Are there any theoretical insights supporting this approach?

**Questions:**

please see the weaknesses

---

> ### Author Response · Authors · 2024-11-20
> **Rebuttal (part 1)**
>
> We thank the reviewer for their thoughtful feedback that has helped us improve our paper. We reply to the points raised below.
>
> > *The motivation for proposing COALA-PG is unclear. It’s not obvious whether the issue is related to variance or other issues when using mini-batches. The authors suggest that larger mini-batches could pose a problem, but this may lead to higher variance in reward summation. However, these points are not extensively discussed in the manuscript. Additionally, compared to M-FOS, it appears that COALA-PG uses the 1/B term to scale rewards, but it seems that this scaling is still related to controlling the policy gradient’s variance.*
>
> The issue in prior work that we are addressing is not related to estimator variance per se, but to incorrect weighting of different elements in the total gradient. Both M-FOS (Eq. 26) and naive policy gradient methods ("batch-unaware”, Eq. 25) incorrectly balance the contributions to the total gradient that stem from the current episode, with those that stem from future episodes. The latter measures the co-player shaping effects of a given action. Although no explicit scaling factor is used for the future inner episode returns, this co-player shaping contribution to the policy gradient scales with O(1/B). This O(1/B) scaling arises because the naive learner averages its parameter update over the B inner episode trajectories, and hence an action in a single trajectory will only have a O(1/B) influence on the naive learner’s parameter update, which leads to an imbalance with the other contribution to the gradient from the current episode. By scaling the inner episode return by 1/B, COALA-PG ensures that both contributions have the same scaling w.r.t. B, and finally by summing the policy gradient over the minibatch instead of averaging, we end up with a policy gradient of O(1). In contrast, M-FOS and batch-unaware COALA do not scale the current inner episode return, which causes the co-player shaping learning signal to vanish w.r.t. the current inner episode, resulting in poor shaping performance. Ultimately, this problem can be traced back to a misspecification of the underlying POMDP, which does not take into account inner episode minibatches. The proposed batched co-player shaping POMDP solves the issue.
>
> To clarify this we have implemented three changes in the paper:
> * We now include a new experimental analysis measuring the present-episode vs. future-shaping contributions to the gradient for different estimators throughout meta-training, see Fig. 12 in App. H.1. COALA-PG balances the two, achieving a ratio of order one, whereas standard (batch-unaware) learning and M-FOS do not.
> * We have included a new visualization of the difference between the COALA-PG update and the batch-unaware policy gradient update (see Fig. 11 in App. G)
> * We have rewritten Theorem 3.1 and the passage below to discuss this point. It now reads:
> > There are three important differences between `COALA-PG` and naively applying policy gradient methods to individual trajectories in a batch. (i) Each gradient term for an individual action $a_l^{i,b}$ takes into account the future inner episode returns averaged over the whole minibatch, instead of the future return along trajectory $b$ (see Fig. 2). This allows taking into account the influence of this action on the parameter update of the naive learner, which influences all trajectories in the minibatch after that update. (ii) Instead of averaging the policy gradients for each trajectory in the batch, `COALA-PG` accumulates (sums) them. This is important, as otherwise the learning signal would vanish in the limit of large minibatches. Intuitively, when a naive learner uses a large minibatch for its updates, the effect of a single action on the naive learner's update is small ($O(1/B)$), and this must be compensated by summing all such small effects. (iii) To ensure a correct balance between the return from the current inner episode $m_l$ and the return from future inner episodes, `COALA-PG` rescales the current episode return by $1/B$. Figure 12 in App. H shows empirically that `COALA-PG` correctly balances the policy gradient terms arising from the current inner episode return versus the future inner episode returns, whereas M-FOS (Lu et al., 2022) and a naive policy gradient that ignores the other parallel trajectories over-emphasize the current inner episode return, causing them to lose the co-player shaping learning signals.
>
> > *Why not use a Q-function instead of summing rewards?*
>
> We now realize that the paper was not clear on this point, so we thank the reviewer for raising this question. We do learn value functions and use them to reduce gradient estimation variance. We have now included a new Appendix section devoted to this (see Appendix A) and we now discuss in the main text the advantage-based estimator that we developed for COALA-PG, and that we use in the simulations.

---

> > ### Author Response · Authors · 2024-11-20
> > **Rebuttal (part 2)**
> >
> > > *In Theorem 3.1, what is the advantage of using batches of inner episode histories instead of a long history? Are there any theoretical insights supporting this approach?*
> >
> > Concatenating in a single long history the multiple inner episodes that the co-player bases its learning on is a viable alternative in theory. In practice, however, it makes the context length of the sequence models prohibitively long, and will thereby make simulations on current hardware much slower. It will also possibly complicate the learning of the sequence model policy, which now needs to manipulate information over much longer timescales.

---

> > > ### Author Response · Authors · 2024-11-25
> > > **Gentle reminder**
> > >
> > > Gentle reminder: The discussion period closes in 2 days. We've tried to address all your concerns with new results, clarifications and an updated manuscript. Please let us know if you have any remaining concerns. We look forward to a productive discussion.

---

> > > > ### Comment · Reviewer_xxyH · 2024-11-26
> > > >
> > > > Thanks for the response of the authors. After reading the comments, I would like to maintain my scores.

---

> > > > > ### Author Response · Authors · 2024-11-26
> > > > >
> > > > > Thank you once again for reviewing our work, and for taking the time to read our rebuttal.
> > > > >
> > > > > We are eager to further improve our paper, and are wondering if you could provide additional guidance on which aspects of it you believe need further strengthening to warrant a higher score. Specifically, are there any remaining concerns from your original review that we haven't adequately addressed, or are there any new concerns that arose after reading our rebuttal?
> > > > >
> > > > > Thank you for your time and we look forward to a productive discussion.

---

> > > > > > ### Comment · Reviewer_xxyH · 2024-12-02
> > > > > >
> > > > > > Thanks for your further comments. I'm not saying that you need to "concatenate" the history, but one could mitigate this problem by using RNNs to maintain hidden states or "belief" states, just like DRQN [1]. This raises my doubt about why we need batches here.
> > > > > >
> > > > > > [1] Matthew J. Hausknecht, Peter Stone. Deep Recurrent Q-Learning for Partially Observable MDPs. AAAI Fall Symposia 2015: 29-37.

---

> > > > > > > ### Author Response · Authors · 2024-12-02
> > > > > > >
> > > > > > > Thank you for taking the time to go through our responses. An RNN can indeed process long sequences by updating a hidden state of fixed size, and in fact, we use Hawk [De et al. 2024], a modern RNN in our experiments. Nevertheless, there remains a significant practical difference between our current batched setup and the setup proposed by the reviewer where we would let the RNN process a single sequence consisting of the concatenated inner episodes in the batch. Below we clarify both setups, making the comparison more explicit.
> > > > > > >
> > > > > > > **Batched setup:**
> > > > > > > * During meta trajectory rollouts, the meta agent's sequence policy sees a single meta trajectory consisting of $M$ concatenated inner episodes.
> > > > > > > * There are $B$ meta trajectory rollouts running in parallel (hence B separate instantiations of the meta agent's sequence policy).
> > > > > > > * Importantly, at each inner episode boundary (i.e. after each T timesteps in the meta trajectory), the naive learner updates its parameters using the current $B$ parallel inner episodes, i.e. the set $(h^b_{mT+1:(m+1)T})_{b=1}^B$, with $m < M$ the index of the current inner episode, and $b$ the index of the parallel trajectory in the batch. See Figure 2 for a schematic of this setup.
> > > > > > >
> > > > > > > **Single trajectory setup:**
> > > > > > > * The naive learner still needs a batch of $B$ inner episode trajectories to do an RL update with low enough variance. As we now want to use a single meta trajectory instead of $B$ meta trajectories in parallel, we still somehow need to have access to $B$ inner episode trajectories.
> > > > > > > * We can solve this in theory by introducing a new type of meta trajectory, where we repeat $B$ times an inner episode in between each update of the naive learner's parameters. Hence, this results in a meta trajectory of length $BMT$ structured as follows: the meta agent plays $B$ inner episodes against the naive learner where the naive learner does not update its parameters, and the naive learner resets its sequence model hidden state at each inner episode boundary; after $B$ inner episodes, the naive learner uses the previous $B$ inner episodes to update its parameters in a batch update (i.e. it restructures the concatenated $B$ inner episodes into a batch); we repeat this procedure $M$ times in a single meta trajectory, resulting in a meta trajectory of length $MBT$.
> > > > > > >
> > > > > > > **Comparison between both setups.**
> > > > > > > * The main practical difference between both setups is that in the batched setup, the meta agent's sequence model needs to process $B$ meta trajectories of length $MT$, whereas in the single trajectory setup, it needs to process a single meta trajectory of length $BMT$. Even though the RNN updates a hidden state of fixed dimension, it still needs to do it in a serial manner in the second setup, whereas in the first setup it can do it in parallel, better leveraging current hardware paradigms. Equally important, training RNNs on long sequences gets harder the longer the sequence is, as the RNN needs to process and combine pieces of information that are further apart. Hence, the difference in trajectory length will lead to significant differences in trainability of the RNNs.
> > > > > > > * Another important difference is that in the batched setup, a meta agent can use the same sequence model when playing against a naive learner or when playing against another meta agent, as both result in sequences of length $MT$. In the second setup, meta agents playing against naive learners would consider trajectories of length $BMT$, and meta agents playing against other meta agents would consider trajectories of length $MT$, hence requiring different learned information processing by the RNN.

---

> > > > > > > > ### Comment · Reviewer_xxyH · 2024-12-02
> > > > > > > >
> > > > > > > > Thanks for your further explanations. I can see the differences between batched updates and single-trajectory updates with the RNN now. Could you also give a hint about when the proposed methods would be effective regarding the number of batches (i.e., batch-size)? When the batch_size is 1, COALA becomes M-FOS. What about a smaller size of batches? Moreover, as you scale the Q-values (compared to M-FOS), will the scaling not affect the variance in policy gradients?

---

> > > > > > > > > ### Author Response · Authors · 2024-12-02
> > > > > > > > >
> > > > > > > > > The reviewer is correct that when the batch_size is equal to 1, COALA-PG becomes equivalent to M-FOS. For batch sizes larger than 1, COALA-PG scales the current batch return by $1/B$, whereas M-FOS does not. This scaling is crucial, as the influence of an action on future inner episodes is of order $O(1/B)$ because the naive learner averages its update over the $B$ inner episode trajectories. Hence, by scaling the inner episode return by $1/B$, COALA-PG ensures that both contributions (current and future episodes) have the same scaling w.r.t. $B$, and finally by summing the policy gradient over the minibatch instead of averaging, we end up with a policy gradient of $O(1)$.
> > > > > > > > >
> > > > > > > > > We cannot see a theoretically justified regime where M-FOS would perform better than COALA-PG, even for small batch sizes, as M-FOS disproportionally weights the current inner episode return, leading to myopic behavior. This is also confirmed by our experimental results, where COALA-PG consistently outperforms M-FOS. Furthermore, in practice, naive learners need significant batch sizes (in our experiments ranging from 8 to 64) in order to have stable learning, due to the variance in policy gradient updates.
> > > > > > > > >
> > > > > > > > > While we did not theoretically investigate the variance resulting from the rescaling, our experimental results showed a significant performance improvement of COALA-PG compared to M-FOS and batch-unaware COALA-PG, indicating that there are no variance issues arising from the rescaling.

---

### Author Response · Authors · 2024-11-20
**General reply to all reviewers**

We thank all reviewers for their useful comments, which helped us improve our paper. Triggered by their questions and concerns, we carried out a number of additional experiments and analyses, and edited the paper to improve its clarity. We summarize the main changes below.

**Detailed proof, restatement and discussion of Theorem 3.1 (COALA policy gradient) + experimental gradient decomposition analysis.** Several reviewers asked us to clarify the differences between COALA-PG, M-FOS, and a naive policy gradient calculation where every batch element is treated independently, and to explain the scaling with minibatch size in more detail. We addressed this recurring issue with a number of changes:
* The appendix now includes a self-contained, detailed derivation of the COALA-PG rule (Appendix D).
* We now present Theorem 3.1 in a form that decomposes current-episode vs. future-episode shaping returns, emphasizing the correct scaling of the two contributions as a function of minibatch size.
* Making use of this decomposition, we expanded the discussion of the theorem, contrasting our rule to M-FOS and naive applications of the policy gradient theorem, which result in improper scaling of each term.
* Lastly, we ran an experimental analysis tracking the relative size of the current-episode vs. future-episode shaping contributions throughout training for COALA-PG, M-FOS, and the naive policy gradient rule. This experiment confirms that while our rule maintains the balance between current-episode and shaping returns, the baseline methods do not (see App. H.1).

**New LOLA baseline.** Reviewer MZki asked us to extend our theoretical link between  COALA-PG and LOLA with experiments with the latter. We now added LOLA results to our experiments, including a 2D version of CleanUp-lite. First, we find that when equipping LOLA with the state-of-the-art Loaded-DiCE higher-derivative estimator [Farquhar et al. 2019], and searching for good hyperparameters, we can make LOLA cooperate on the iterated prisoner’s dilemma with sequence model policies, in contrast to what previous findings suggest [Zhao et al. 2022]. When scaling to the more complex CleanUp-lite-2D environment (see below), the LOLA baseline is outperformed by our COALA-PG estimator, suggesting that the model-free COALA-PG approach is more suitable to scale to larger environments (see Fig. 13 in App. H.2).

**CleanUp-lite-2D.** As requested by multiple reviewers, we increased the difficulty of our simplified CleanUp environment by turning it into a 2D grid world and show that COALA-PG agents find cooperative solutions, in contrast to M-FOS and LOLA (see App. H2). The Appendix now also contains an extended discussion of the environment (see App. B.1). We are still running some additional baselines on this new environment, and will move it from the appendix to the main manuscript once those are done.

**Analysis of the emergence of cooperation with LOLA.** The new version of the paper includes a novel analysis of the emergence of cooperation between agents trained with the LOLA algorithm. Our analysis reveals a crucial role played by a hyperparameter of the algorithm, namely the number of simulated co-player learning steps (often referred to as ``look-ahead steps”). As this number increases, LOLA agents transition from cooperation to extortion strategies resulting in mutual defection. Our analysis explains the origin of this phenomenon, and the similarities and differences between LOLA and COALA.

**Extended results on the analytical iterated prisoner’s dilemma.** Encouraged by Reviewer UL6h, we added new results on the analytical iterated prisoner’s dilemma, showing that meta agents learn extortion strategies similar to the Zero-Determinant extortion strategies first discovered by Dyson & Press (2012), and that unconditional defection is not a Nash equilibrium anymore in the mixed group training setup, allowing meta agents to escape (see App. C.1).

**Advantage/value estimation.** Upon request of reviewer xxyH, we added an Appendix section describing in detail the value and advantage function estimators we developed, and how COALA-PG can be used in combination with the standard A2C and PPO reinforcement learning algorithms.

**Improved figure clarity.** All experimental figures were decluttered, and replotted with larger line and font sizes.

**Updated manuscript.** We included all mentioned changes in an updated version of the manuscript.

---

### Author Response · Authors · 2024-12-02
**Final call for action**

A gentle reminder that today is the last day for the reviewers to respond to our rebuttals. To the best of our knowledge, we have addressed all the main concerns and comments raised during the reviews and the discussion period with new results and clarifications, which we have added to our manuscript.

These improvements have been acknowledged by reviewer pFwY, who maintained their score of 8, and they led reviewers MZki to raise from 5 to 8 and fP5Y to raise from 3 to 6. To name a few highlights, these reviewers praised the paper's 'extensive theoretical comparisons to prior work' and 'interesting findings', 'excellent presentation', 'sound improvements over current opponent shaping methods with self-interested learners', the 'introduction of the first unbiased, higher-derivative-free policy gradient algorithm for learning-aware RL, a significant contribution', 'the proposal of a novel explanation for cooperation', and that the paper 'tackles a core issue in multi-agent learning, fostering cooperation among self-interested agents'.

We kindly ask reviewers **xxyH, GXnN, fP5Y and UL6h** to respond whether there are any remaining concerns that we haven't adequately addressed, or new concerns that arose after reading our rebuttal, that would prevent the reviewers from further raising their score.

Below we briefly summarize the main concerns of reviewers xxyH, GXnN, fP5Y and UL6h and how we addressed them with new results and clarifications.

**_Reviewer xxyH_**

**Main concerns and comments:**
* **Unclear motivation for COALA-PG:** The reviewer questioned the specific issue that COALA-PG addresses, asking whether it's related to variance or other problems arising from the use of mini-batches.
* **Comparison to Q-function:** The reviewer asked why we didn't use a Q-function instead of summing rewards.
* **Batching versus concatenation:** The reviewer asked why we use batches of inner episode histories instead of concatenating inner episodes into a single long history.

**Our responses:**
* **Clarified motivation:** We explained that the issue is not about estimator variance but about incorrect weighting of gradient contributions from current and future episodes in prior work. We provided a detailed explanation of how COALA-PG correctly balances these contributions, which we supported with a new experimental analysis (Figure 12 in Appendix H.1). We also included a new visualization of the difference between the COALA-PG update and the batch-unaware policy gradient update (Figure 11 in Appendix G).
* **Comparison to Q-function:** We acknowledged that the original paper lacked clarity on this point and added a new Appendix section (Appendix A) detailing the value and advantage function estimators we developed, including the advantage-based estimator used with COALA-PG.
* **Batching versus concatenation:** We clarified the practical advantages of using batches, such as avoiding prohibitively long sequence model contexts and simplifying the learning of the meta agent.
The reviewer acknowledged reading our responses, but maintains their score, without providing any explanation on which further concerns remain that prevent them from further raising their score.

**_Reviewer GXnN_**

**Main concerns and comments:**
* **Partial Observability:** The reviewer was concerned that full history dependency might reduce the applicability of our framework to truly partially observable scenarios.
* **Simplistic Environments:** The reviewer found the experimental scenarios too simplistic and not fully demonstrative of the framework's ability to handle complex, dynamic cooperation in partially observable environments.
* **Lack of Convergence Analysis:** The reviewer pointed out the absence of a formal convergence analysis for COALA-PG in non-stationary environments.

**Our responses:**
* **Partial Observability:** We addressed this by (i) including a new proof of Theorem 3.1, showing that COALA-PG remains valid in partially observable settings (Appendix D), and (ii) introducing a more challenging 2D version of the CleanUp-lite environment (Appendix B.1, H.2).
* **Simplistic Environments:** We added results on the new, more complex CleanUp-lite-2D environment, demonstrating the effectiveness of COALA-PG in a harder partially observable setting (Appendix B.1, H.2).
* **Lack of Convergence Analysis:** We responded by (i) providing a detailed empirical analysis of learning trajectories on the iterated prisoner's dilemma (Section 4), (ii) adding new results on the convergence behavior of meta-agents (Appendix C.1), and (iii) highlighting the non-stationary nature of our experiments in Section 5 (Figure 5C and Figure 7)

---

> ### Author Response · Authors · 2024-12-02
> **Final call for action (part 2)**
>
> **_Reviewer fP5Y_**
>
> **Main concerns and comments:**
> * **Novelty of the batched co-player shaping POMDP.** The reviewer raised concerns about the similarity between the batched co-player shaping POMDP, and a standard parallel-environment-POMDP setup.
> * **CleanUp-lite environment specifications.** The reviewer asked for more information on the CleanUp-lite environment and how it differs from the CleanUp environment in Melting Pot.
> * **Comparison to reward-shaping methods.** The reviewer asked why we do not compare our method against established reward-shaping methods.
>
> **Our responses and further discussion.**
> * **Novelty of the batched co-player shaping POMDP.** We clarified that in our batched co-player shaping POMDP, parallel trajectories interact with each other through the naive learners’ parameter updates, which use the full batch of current inner episodes. Correctly formulating this batched co-player shaping is crucial for obtaining a correct policy gradient, explaining the improved performance of COALA-PG compared to M-FOS which uses a biased policy gradient estimator. We included a new schematic (Figure 11) to clarify our setup.
> * **CleanUp-lite environment specifications.** We added detailed environment specifications in Appendix B.1.
> * **Comparison to reward-shaping methods.** We clarified that there are two main approaches in the mixed-motive multi-agent RL (MARL) literature to solve mixed-motive games: (i) learning-aware RL methods, and (ii) reward-shaping to change the incentive structure of the game, making cooperative equilibria easier to find. Our method is situated in the former field, motivating our choice of comparing it to other learning-aware MARL methods. To satisfy the reviewer’s request for more comparisons, we included an extra baseline: LOLA equipped with the state-of-the-art Loaded DiCE estimator.
> * The reviewer acknowledged that we addressed most of their questions, and raised their score from 3 to 6.
> * The reviewer raised a last question that they would still be curious about how the performance of our method compares to other reward-shaping methods. We clarified that the main motivation for our work (and the learning-awareness field in general) is fundamentally different from the reward-shaping field. We focus on how cooperation can be achieved from optimizing purely selfish objectives, without requiring privileged access to other agent’s rewards and without relying on all co-player to use pro-social reward-shaping. Hence, comparing to reward-shaping methods that change purely selfish objectives to pro-social objectives, would not fit well in our current work. We note that Reviewer MZki stressed the ‘sound improvements’ brought by our method in the most general setting, where no explicit assumptions need be made on the other agents’ learning algorithms and objective functions, thus covering the case of purely self-interested agents.

---

> > ### Author Response · Authors · 2024-12-02
> > **Final call for action (Part 3)**
> >
> > **_Reviewer UL6h_**
> >
> > **Main concerns and comments.**
> > * **Ambiguous metrics:** the reviewer was concerned that we did not sufficiently discuss which metrics we use to compare COALA-PG to other methods, and how these metrics are reflected in the design of our method.
> > * **Co-player policy estimation:** the reviewer asked for a clarification on how meta agents can estimate the co-players’ policy and whether meta agents have explicit access to the co-players’ parameters through an extra input.
> > * **Proof of Theorem 3.1:** The reviewer raised concerns about the correctness of the proof, as we leveraged Sutton’s policy gradient theorem which applies to MDPs, but considered a POMDP setting for our COALA-PG method.
> > * **Extortion in IPD:** The reviewer asked whether an extortion policy in our analytical iterated prisoner’s dilemma (c.f. Figure 3) can reach higher returns than mutual cooperation.
> >
> > **Our responses and further discussion with the reviewer.**
> >
> > * **Ambiguous metrics:** We acknowledged that the first version of our manuscript did not sufficiently discuss the used metrics, and addressed this in the new version of the manuscript by updating the relevant figure captions, and by clearly stating in Section 5 that our aims are twofold: to understand (i) when meta-agents succeed in exploiting naive agents, and (ii) when cooperation is achieved among meta-agents.
> > * In a subsequent response, Reviewer UL6h acknowledged that our metrics are consistent. The reviewer raised further concerns that (i)  “exploiting naive agents” is only a marginal point, (ii) that while cooperation among meta agents (social welfare) is a relevant metric, this metric is not used as an explicit optimization objective in our method, and (iii) asked why optimizing selfish objectives could lead to outcomes that are more Pareto-dominant.
> > * As a response, we clarified that the main contribution of our paper is showing how cooperation among meta agents can arise from optimizing selfish objectives. (ii) Hence, using selfish objectives for agents instead of the pro-social objective of social welfare is crucial for our scientific investigation. (i) We clarified that while co-player shaping (extorting naive learners) is indeed only an instrumental goal, this part is crucial to obtain cooperation among meta agents, justifying why we need a detailed analysis of the co-player shaping capabilities of COALA-PG compared to other state-of-the-art methods. (iii) We clarified that both Section 4 and a new appendix section C.1 are dedicated to the question of why optimizing selfish objectives could lead to outcomes that are more Pareto-dominant. We note that the ‘sound improvements’ brought by our method to the case of purely self-interested learners was highlighted by reviewer MZki. This problem was further cited as a  ‘core issue in multi-agent learning’, and quoted as a strength of our paper by reviewer fP5Y.
> > * **Co-player policy estimation:** we clarified that the meta agent’s policy does not require co-players’ parameters as explicit inputs, and that instead the sequence model policy meta-learns to infer “in-context” the parameters and learning algorithms employed by co-players based solely on observations including their past actions. Reviewers GXnN and MZki explicitly acknowledged this aspect of our method as one of its key strengths.
> > * The reviewer asked a follow-up question on which policy of the naive learner is estimated by the meta agent, as the naive learner’s policy is updated multiple times during a single meta trajectory. We clarified that the in-context inference of the meta-agent’s sequence model implicitly estimates the current policy for a considered timestep $l$ in the meta trajectory, akin to e.g. Kalman filtering.
> > * **Proof of Theorem 3.1:** While the first version of our proof was correct when taking into account that one can transform any POMDP to a belief-state MDP, we included a new proof in Appendix D that does not rely on Sutton’s policy gradient theorem, providing a more direct line of arguments to proof Theorem 3.1. The reviewer raised further concerns that it remained unclear to the reviewer which policy of the naive learner is being estimated (c.f. previous question). We clarified this (c.f. previous question) and noted that this does not impact the proof. We note that reviewers xxyH and pFwY praised the completeness, soundness and the extent of our theoretical results. Moreover, the unbiased policy gradient rule was in particular highlighted as a strength by reviewers fP5Y, MZki and GXnN.
> > * **Extortion in IPD.** We clarified that Figure 3 shows the average return over the whole meta trajectory, hence even though extortion policies successfully extort naive learners at the end of the meta trajectory, the overall return can be below 1. We redid the experiments of Figure 3 with longer meta trajectories, making it clear that the extortion policies achieve a higher return than mutual cooperation.

---

### Meta-Review · Area_Chair_X5cM · 2024-12-19

**Metareview:**

The reviewers acknowledged that the paper studies an interesting problem of cooperation among self-interested learning agents in multi-agent systems,  and the proposed method of learning-aware policy gradients is novel with significant contributions. However, the reviewers also raised several concerns and questions in their initial reviews. We want to thank the authors for their responses and active engagement during the discussion phase. The reviewers appreciated the responses and have an overall positive assessment of the paper for acceptance. The reviewers have provided detailed feedback, and we strongly encourage the authors to incorporate this feedback when preparing the final version of the paper.

**Additional Comments On Reviewer Discussion:**

The reviewers raised several concerns and questions in their initial reviews. After the discussion, most (five out of six) reviewers have a positive assessment of the paper. One of the reviewers has a borderline rating of 5 – the authors have provided a detailed response to address the reviewer's concerns; however, this reviewer didn't participate in the discussions.

---

### Decision · Program_Chairs · 2025-01-22

Accept (Poster)